# Regenerative and Anti-Inflammatory Potential of Regularly Fed, Starved Cells and Extracellular Vesicles In Vivo

**DOI:** 10.3390/cells11172696

**Published:** 2022-08-30

**Authors:** Federico Ferro, Renza Spelat, Georgina Shaw, Cynthia M. Coleman, Xi Zhe Chen, David Connolly, Elisabetta M. F. Palamá, Chiara Gentili, Paolo Contessotto, Mary J. Murphy

**Affiliations:** 1Department of Medical, Surgery and Health Sciences, University of Trieste, 34125 Trieste, Italy; 2College of Medicine, Nursing and Health Science, School of Medicine, Regenerative Medicine Institute (REMEDI), National University of Ireland Galway (NUI Galway), H91 W2T9 Galway, Ireland; 3Neurobiology Sector, International School for Advanced Studies (SISSA), 34136 Trieste, Italy; 4Discipline of Biomedical Engineering, School of Engineering, College of Science and Engineering, National University of Ireland, H91 TK33 Galway, Ireland; 5Department of Experimental Medicine (DIMES), University of Genoa, 16132 Genoa, Italy; 6Department of Molecular Medicine, University of Padova, 35122 Padova, Italy

**Keywords:** mesenchymal stem/stromal cells, bone micro-computed, diabetes, inflammation, injury/fracture healing, starvation tomography (μCT), extracellular vesicles

## Abstract

Background: Mesenchymal stem/stromal cells (MSC) have been employed successfully in immunotherapy and regenerative medicine, but their therapeutic potential is reduced considerably by the ischemic environment that exists after transplantation. The assumption that preconditioning MSC to promote quiescence may result in increased survival and regenerative potential upon transplantation is gaining popularity. Methods: The purpose of this work was to evaluate the anti-inflammatory and regenerative effects of human bone marrow MSC (hBM-MSC) and their extracellular vesicles (EVs) grown and isolated in a serum-free medium, as compared to starved hBM-MSC (preconditioned) in streptozotocin-induced diabetic fractured male C57BL/6J mice. Results: Blood samples taken four hours and five days after injection revealed that cells, whether starved or not, generated similar plasma levels of inflammatory-related cytokines but lower levels than animals treated with EVs. Nonetheless, starved cells prompted the highest production of IL-17, IL-6, IL-13, eotaxin and keratinocyte-derived chemokines and induced an earlier soft callus formation and mineralization of the fracture site compared to EVs and regularly fed cells five days after administration. Conclusions: Preconditioning may be crucial for refining and defining new criteria for future MSC therapies. Additionally, the elucidation of mechanisms underpinning an MSC’s survival/adaptive processes may result in increased cell survival and enhanced therapeutic efficacy following transplantation.

## 1. Introduction

The healing of bone fractures produced by severe trauma remains a problem for contemporary medicine and comes at a high worldwide cost. Furthermore, aging and age-related chronic diseases such as diabetes exacerbate the condition, resulting in decreased osteogenic differentiation and bone quantity, directly impacting bone healing ability in affected individuals [1]. Bone healing is a dynamic process that includes inflammation, granulation tissue growth, intramembranous and endochondral ossification and, finally, remodeling of the resulting woven bone to a lamellar structure until the original shape is restored. The healing process is orchestrated by a complicated sequence of inflammatory, angiogenic, osteo/anabolic and osteo/catabolic mediators, which are only partially understood [2]. Adult mesenchymal stem/stromal cells, acknowledged as the primary cell source for regenerative medicine therapy, can be readily isolated and retain the capacity to proliferate ex vivo [3]. Mesenchymal stem/stromal cell (MSC) and derived extracellular vesicle (EVs) transplantation have been viewed as viable treatments in regenerative medicine due to their multilineage differentiation and paracrine benefits [4,5,6]. When in vivo transplanted or injected MSC are exposed to an ischemic environment, the nutritional restriction and low oxygen content significantly lower their lifespan, via apoptotic processes [7], and their therapeutic potential [5,8,9,10]. Because of their low viability after transplantation, it is currently assumed that the favorable benefits of MSC, such as enhanced angiogenesis, inflammation suppression and immune-modulation, are primarily due to released trophic factors and to EVs, or linked to apoptotic processes [7,11], rather than their direct regenerative capabilities [12].

In vivo mesenchymal stem/stromal cells typically reside in microenvironments known as stem/stromal cell niches, which have been shown to preserve cell stemness and self-renewal by maintaining cell quiescence through the lack of nutrients and oxygen [13,14]. When MSC are cultured in vitro, their glucose and oxygen consumption rise considerably, resulting in poor adaptability and a shorter lifespan when transplanted [15,16,17,18]. As a result, approaches capable of enhancing the MSC’s survival in vivo, such as promoting quiescence or decreasing metabolic demands, are of great interest in an effort to maximize their therapeutic efficacy and completely realize their potential [17,19]. Many studies have applied “preconditioning” strategies in vitro to mimic the in vivo context [10,16,18,19,20]. Similarly, we discovered that preconditioning human bone marrow mesenchymal stem/stromal cells (hBM-MSC) by inducing a complete starvation in a serum-free medium (SFM) was more appropriate than in a fetal bovine serum (FBS)-containing medium [17]. In keeping with a previous study [10], we established that hBM-MSC may employ gluconeogenesis, β-oxidation and autophagy to overcome the poor metabolic conditions established under long-term starvation [17]. Furthermore, Moya et al. (2017) demonstrated that induced quiescence, as well as the associated lower energy requirement achieved by a two-day serum deprivation prior to cell transplantation, protects hBM-MSC from the abrupt transitions from in vitro to the harmful ischemic environment in vivo [10].

Therefore, the concept of priming cells by preconditioning them to promote quiescence is gaining popularity as a way to improve the anti-inflammatory, regenerative and survival rates of MSC in vivo [5,10,11,17,19].

According to this theory, preconditioned MSC will be a more suitable method for implementing mesenchymal cell therapies and tissue engineering applications because of their ability to produce trophic factors that improve angiogenesis, reduce inflammation, favor bone regeneration and increase the resident stem/stromal cells’ recruitment and differentiation.

Diabetes is a disease that triggers chronic hyper-inflammatory conditions [1,21,22,23], impairing the activity and presence of neutrophils [1], monocytes and dendritic cells [24] that result in pro-inflammatory and chemotactic mediators’, such as IL-6, MCP-1 (CCL2), IL-1, TNF-α, IL-4 and IFN-γ, persistent secretion [25,26,27,28,29]. All these conditions contribute to induce anti-osteogenic activity, decreased soft callus formation and enhanced osteoclastic and bone turnover activity [30], thereby delaying fracture healing [1,22,31].

Such a hyper-inflammatory milieu is recapitulated in the diabetic STZ-induced animal models used in this research, as well as by McCauley J. et al. [32], and it lays at the basis of the reason as to why we chose this fracture model to test our hypothesis.

As a consequence, in this study we compared the effects of a prolonged nutritional starvation (preconditioning) period for regularly fed MSC and their matched EVs on the regenerative and anti-inflammatory properties of hBM-MSC in a diabetic fractured repair model.

## 2. Materials and Methods

### 2.1. MSC Isolation from Human Bone Marrow

Human bone marrow samples from the iliac crest of four healthy donors were used to isolate and expand in vitro hBM-MSC. The BM was taken from donors after informed consent was obtained in accordance with Galway University Hospital’s requirements under protocol Reference 02/08, approved by the institutional Clinical Research Ethical Committee. Mononuclear cells were counted using a hemocytometer and isolated from the BM by directly plating them into T175 flasks (Corning, NY, USA) at a density of 2.25–3 × 10^5^/cm^2^ in a chemically defined serum-free medium (SFM) (https://patents.google.com/patent/WO2015121471A1) composed of α-minimum essential medium (α-MEM, Gibco, Grand Island, NY, USA), lipoprotein 40 μg (Sigma-Aldrich, Saint Louis, MO, USA), dexamethasone 50 nM (Sigma-Aldrich), ascorbic acid 2-phosphate 100 μM (Sigma-Aldrich), human serum albumin 1% (Baxter, Deerfield, IL, USA), insulin-transferrin-selenium 1% (ITS, BD Biosciences, Oxford, CA, USA), transforming growth factor β (TGF-β) 20 ng/mL and fibroblast growth factor 2 (FGF-2) 10 ng/mL (Thermo Fisher Scientific, Waltham, MA, USA). Cells were cultured at 37 °C, 5% CO_2_ until passage 3 (P3) under normoxia (21% O_2_) with a regular twice weekly medium change and cryopreserved in liquid nitrogen. To obtain the desired number of cells that had to be injected, hBM-MSC were thawed around 80 days before transplantation and cultured until P6, compensating for a cell death ratio of 15–20 percent during the subsequent starvation. Following P6, “starving” hBM-MSC were left in the same medium with no exposure to fresh medium for 70 days under normoxia. After that period, starving hBM-MSC were washed in phosphate-buffered saline (PBS, Gibco), detached using 0.25% trypsin (Gibco) for 15 min, counted and resuspended in PBS at 5000 cells/µL. Regularly fed cells were cultured in SFM until P3 and cryopreserved at 5000 cells/µL in liquid nitrogen until in vivo injection.

### 2.2. Extracellular Vesicle Isolation

EVs were separated from P3 regularly fed cells as follows: sub-confluent hBM-MSC cultured in SFM were washed three times in PBS and cultured in basal medium (αMEM, Gibco) for 72 h. Conditioned medium (CM) was harvested after 72 h and processed immediately through two subsequent centrifugation steps [33]. This clarified CM was differentially ultra-centrifuged (Optima XPN-100, Beckman-Coulter, Brea, CA, USA) at 10,000× *g* for 40 min at 4 °C and, subsequently, twice at 100,000× *g* for 2 h at 4 °C to pellet EVs [33]. EVs were resuspended in PBS and stored at −80 °C for further experiments or until in vivo injection.

### 2.3. Nanoparticle Tracking Analysis

EV samples were characterized and quantified by Zetaview (Particle Metrix GmbH, Inning am Ammersee, Germany) nanoparticle tracking analysis (NTA), equipped with a sample cell and two lasers (488 nm and 640 nm) and Zetaview 8.05.14_SP7 software. After calibration with 100 nm polystyrene beads, samples were diluted in filtered PBS and injected into the sample cell using a 1 mL syringe. Size distribution analyses of 11 different positions were performed for each sample on at least three different EVs preparations.

### 2.4. Transmission Electron Microscopy

EV samples were fixed in 2% paraformaldehyde and adsorbed for 10 min to formvar–carbon-coated copper grids. Grids were negatively stained with 2% uranyl acetate for five min at room temperature (RT). Stained grids were embedded in 2.5% methylcellulose for improved preservation and air dried before examination. Electron micrographs were taken with a transmission electron microscope (TEM) (HT7800 series, Tokyo, Japan) equipped with Megaview 3 digital camera and Radius software (EMSIS, Münster, Germany).

### 2.5. Immunoblot

Isolated EVs were resuspended in RIPA buffer (1% Nonidet p-40, 0.1% SDS, 0.1% sodium deoxycholate, protease inhibitor cocktail (Sigma-Aldrich), in PBS pH 7.5) and 2 μg of proteins for each sample were loaded on 4–12% NuPAGE Bis-Tris gel (Life Technologies, Carlsbad, CA, USA). Electrophoresis was performed and proteins were blotted on a polyvinylidene fluoride (PVDF) membrane (Millipore, Burlington, MA, USA). The membrane was incubated overnight at 4 °C with specific primary antibodies for: anti-human CD9 (1:1000 dilution, Abcam, Cambridge, UK), anti-human CD63 (1:1000 dilution, Thermo Fisher Scientific, Waltham, MA, USA), anti-human CD81 (1:5000 dilution, BD Biosciences, San Jose, CA, USA), anti-human syntenin-1 (1:1000 dilution, Abcam), anti-flotillin-1 (1:10,000 dilution, Abcam). A specific HRP-conjugated secondary antibody (1:2000 dilution, Cell Signaling Technology, Danvers, MA, USA) was used for the detection. Positivity was highlighted by providing the substrates for the chemiluminescence reaction of HRP (GE Healthcare, Chicago, IL, USA). Gel running was performed under non-denaturizing conditions for the detection of CD9, CD81 and CD63 and under denatured conditions for other markers.

### 2.6. Flow Cytometry

Starved P6 hBM-MSC at 70 days, as well as the same starting population at P6, and regularly fed hBM-MSC at P3, were stained with the following antibodies as per manufacturer’s instructions: CD105 (Invitrogen, Carlsbad, CA, USA), CD73, CD90, CD34, CD45, CD106, CD146 (BD) and CD271 (Miltenyi Biotec, Bergisch Gladbach, DE), HLA-DR, IgG1 BD, IgG2b and IgG1κ (Invitrogen). Briefly, 2 × 10^5^ cells were incubated at 4 °C for 45 min with the antibody dilutions (1:50–1:200) and washed in PBS before analysis by flow cytometry. EVs were characterized using a non-conventional flow cytometry approach [34]. EVs were suspended in filtered PBS/2 mM EDTA (1 × 10^9^ particles in 100 μL) and stained with 1 μM CFDA-SE Cell Tracer Kit (VybrantTM, Waltham, MA, USA) at RT. A 4 °C control was also carried out to verify staining specificity. A mixture of fluorescent beads with Megamix-Plus FSC and Megamix-Plus SSC (Biocytex, Marseille, France) was used to discriminate EVs’ size. Upon accurate titration, expressions of typical vesicle markers CD9 (Biolegend, San Diego, CA, USA), CD63 and CD81 (BD) were evaluated within the CFDA-SE positive events. The assessments were executed using FACS Canto (BD) and FACSAria II (BD) cytometers and FlowJo (BD) software for post-analysis as previously described [17].

### 2.7. Differentiation Assays

Starved hBM-MSC at P6 and regularly fed P3 cells from the same donor were differentiated in triplicate through the osteoblastic, adipocyte, and chondroblastic lineages as described previously [17]. Osteogenesis and adipogenesis of the hBM-MSC were assessed using Alizarin Red S (Sigma) at day 17 for osteogenesis and Oil Red O (Sigma) at day 19 for adipogenesis. Chondrogenesis was assessed by Safranin O and Alcian Blue (pH 2.5) staining; dimethyl methylene blue (DMMB) assays, followed by DNA quantification with PicoGreen (Molecular Probes, Eugene, OR, USA) at 21 days, were used to quantify glycosaminoglycans.

### 2.8. Protein Array

The C-Series human growth factor antibody array C1 was used to compare 1 mL each of SFM recovered after starvation and fresh SFM (RayBio, Peachtree Corners, GA, USA) in duplicate. X-ray films were then used to detect horseradish peroxidase signals and quantified with Fiji ImageJ software (version 1.51r; NIH, Bethesda, MD, USA).

### 2.9. Fracture Model

For the in vivo studies, regularly fed (P3) and starved hBM-MSC (P6) and EVs from regularly fed cells (P3) were tested in vivo for the ability to mediate repair of a femoral fracture in a diabetic mouse model. The local National University of Ireland Galway Animal Care & Research Ethical Committee granted full ethical permission (Application 17-Oct-01) for this study with a license issued by the Irish National Health Products Regulatory Authority (HPRA) under project authorization number AE19125/P075.

Briefly, male C57BL/6J mice were obtained from Charles River laboratories, UK, at six weeks old. Nine week old mice were intraperitoneally injected with streptozotocin (STZ; 50 mg/kg, Sigma) in Hank’s balanced salt solution (HBSS, Sigma) to induce the diabetic condition. Injections were administered once a day for five days, and control animals were injected with the same volume of HBSS. Blood samples were obtained before the STZ injection to test baseline glucose levels, as well as weekly for three weeks to determine the onset and persistence of the diabetic state. All animals with no signs of pain and a glucose level ≥ 13 mM for three consecutive weeks were classified as hyperglycemic and enrolled in the study. After three weeks (day 0) animals were anesthetized with isoflurane, the joint surface was exposed, and an intramedullary stabilizing pin, a 27 G syringe needle, was inserted into the marrow cavity and confirmed using an X-ray machine, as previously described [21]. Mice femurs were positioned on the fracture device [35], a three-point bending guillotine, under the center of the sliding weight before being dropped onto the leg, causing the fracture [21]. An X-ray was taken immediately after the fracture, to ensure that the location of the intramedullary pin had not altered and that the transverse fracture had occurred. Following anesthesia, animals were given 0.05 mg/kg buprenorphine and were separately housed [21].

### 2.10. Cells and EVs Administration

Under isoflurane anesthesia, cells or EVs were administered on day two post fracture and randomized to the treatment group; the precise position of the injection into the fracture site was confirmed using an X-ray apparatus. A single dosage of starved and regularly fed hBM-MSC (2.5 × 10^5^ in 50 µL in PBS, Gibco) was tested, as were EVs produced from the same number of regularly fed cells and supplied through local injection at the fracture site. Mineralized material was monitored radiographically at specified time-points (day seven: *n* = 5 to 6 per group and day 23: *n* = 6 to 8 per group). Control animals were injected with the same volume of PBS (day seven: *n* = 5 per group and day 23: *n* = 6 to 8 per group).

### 2.11. Animal Sacrifice and Tissues Collection

Animals were sacrificed by CO_2_ inhalation and cervical dislocation on the seventh and 23rd days post fracture. Muscle and soft tissues were carefully removed from harvested femurs before wrapping the bones in PBS-soaked gauze (Gibco). Blood samples were collected from the tail vein in 0.5 M EDTA (Gibco) and rapidly frozen at −20 °C. Samples were taken on day two, four hours after injection, and day seven post fracture from diabetic and non-diabetic animals, injected and non-injected with hBM-MSC and EVs.

### 2.12. Multiplex ELISA Cytokine Analysis

Post-traumatic inflammation was measured using a multiplex enzyme-linked immunosorbent assay (ELISA) in diabetic animals treated with hBM-MSC regularly fed, starved, and EVs derived from hBM-MSC; diabetic fractured and non-diabetic fractured animals were used as sham (n = 3 to 6). Plasma concentrations of interleukin (IL)-1α, IL-1β, IL-2, IL-3, IL-4, IL-5, IL-6, IL-9, IL-10, IL-12(p40), IL-12(p70), IL-13, Il-17, eotaxin, granulocyte colony-stimulating factor (G-CSF), granulocyte-macrophage colony-stimulating factor (GM-CSF), interferon-γ(IFN-γ), keratinocyte-derived chemokine (KC, CXCL1), monocyte chemotactic protein 1(MCP-1), macrophage inflammatory protein-1α, 1β (MIP-1α and MIP-1β), regulated on activation, normal T cell expressed and secreted (RANTES) and tumor necrosis factor-α (TNF-α) were quantified using 50 μL of blood and the Bio-Plex Pro Mouse Cytokine (Bio-Rad Laboratories, Hercules, CA, USA). Data were analysed with the Luminex 200 Total System (Bio-Rad).

### 2.13. Micro-Computed Tomography

Micro-computed tomography (μCT) imaging was performed using a Scanco μCT100 (SCANCO Medical AG, Brüttisellen, Switzerland) with mineral matrix samples scanned in PBS within a 15 mm sample holder. A sample length of 6.78 mm spanning the fracture area was selected and scanned within a 15.1 mm field of view, with an X-ray power of 70 KeV through a 0.1 mm aluminum beam hardening filter and an integration time of 560 ms generating scan data with voxel size of 7.4 μM.

Weekly calibration checks were carried out by scanning a phantom containing five rods of varying known mineral densities (0, 100, 200, 400 and 800 mg HA/ccm) and three coordinate calibration pins for x, y and z calibration. All quality control checks during this study indicated that the accuracy of the density and coordinate calibrations were within the recommended range provided by Scanco to denote appropriate instrument calibration. Slice data were reconstructed automatically using Scanco density, x, y and z coordinate algorithms and a cylindrical volume of interest within each scan volume was defined for 3D evaluation. New mineral matrix calluses and original bone in the μCT slices were separated from other components present in the sample (i.e., marrow and PBS) using threshold values of 123.8 mg HA/ccm for new mineral matrix and 430.6 mg HA/ccm for original bone to create 3D reconstructions of the fractured regions. Mineral matrix regeneration was quantified using the Fiji ImageJ and the BoneJ plug-in [36]; briefly, μCT images were cropped to the desired size, binarized and analysed.

### 2.14. Statistical Analysis

Experiments were conducted at least in triplicate and repeated twice at a minimum. Data were compared using SPSS software version 22.0 (IBM, New York, NY, USA). All results are presented as the mean ± SD. Animal sample size was n = three to sixteen per group and by time-point. Statistical significance was evaluated by Student’s *t*-test, one-way analysis of variance (ANOVA) followed by Tukey’s, Fisher’s or Bonferroni’s post hoc test, as appropriate. Paired *t*-test was used to compare samples taken from related groups. Significance was set to *p* ≤ 0.05 *, +, **.

## 3. Results

### 3.1. Starved and Regularly Fed Cells Characterization

Regularly fed hBM-MSC used in this study were isolated from fresh human BM and cultured in a serum-free, xenogen-free chemically defined culture medium until P3, and then stored in liquid nitrogen until in vivo injection. Prior to transplantation, a parallel aliquot of P3 hBM-MSC was thawed, allowed to expand until P6 and exposed to total starvation for 70 days under normoxic conditions in order to produce the required number of cells to be injected (Figure 1A). Starvation led to a considerable reduction in cell number (about 15 to 20%) and different morphological and functional changes compared to the control hBM-MSC. By day 70, a more homogeneous population of fibroblast-like cells, finely interconnected, was present in the culture [17] (Figure 1B).

Flow cytometry was used to assess the expression of the cluster of differentiation (CD) markers in cells from each condition. There were no variations in the expression of CD34 (≈0–1%), CD45 (≈0–1%), HLA-DR (≈0–1%), CD362 (≈3–10%), CD106 (≈0.4–1.5%), CD271 (≈0.6–11.5%), CD73 (≈97–100%), CD90 (≈98–100%) and CD105 (≈99–100%) in P6 starved, P3 regularly fed and the equivalent hBM-MSC at P6 before the starvation period. By contrast, CD146 was significantly reduced as a result of the starvation process (50 ± 7.1%) as compared to the same cells at P6 before starvation (77.5 ± 17.4%), and the regularly fed cells (88.6 ± 4.62) (*, ** *p* ≤ 0.05) (Figure 1C).

Thereafter, P6-starved cells were tested for their differentiation potential and induced through the adipocyte, chondroblastic and osteoblastic lineages. Results showed that starved hBM-MSC retained their osteoblastic and chondroblastic differentiation potential and increased their adipogenic potential (0.79 ± 0.03 fold change (F.C.)) as compared to the regularly fed cells at P3 (0.51 ± 0.18 F.C.) (*p* ≤ 0.05) (Figure 2A–D).

Starved cells at the end of the 70 day in vitro starvation period secreted growth factors in the culture medium, including insulin growth factor binding protein 6 (IGFBP6) (1.49 ± 0.08 F.C.), IGFBP3 (1 ± 0.21 F.C.), IGFBP2 (7.75 ± 0.001 F.C.), hepatocyte growth factor (HGF) (5.52 ± 0.02 F.C.), insulin growth factor 2 (IGF2) (1.01 ± 0.07 F.C.), platelet-derived growth factor α (PDGFα) (3.31 ± 0.05 F.C.), transforming growth factor 2 (TGFβ2) (2.8 ± 0.24 F.C.) (*p* ≤ 0.05) and vascular endothelial growth factor receptor 2 (VEGFR2) (1.05 ± 0.01 F.C.), which were undetectable in fresh medium (Figure 2E).

### 3.2. EVs Characterization

EVs from P3 regularly fed cells were characterized, according to MISEV2018 [37]. TEM analysis on the isolated vesicles was performed to check the EVs’ morphology (Figure 3A) showing morphological and dimensional heterogeneity. A wide-ranging size distribution was also confirmed by NTA, displaying a mean EV size of 133.6 ± 4.74 nm (Figure 3B). The expressions of EV surface markers CD63 and CD81 and additional vesicular markers syntenin-1 and flotillin-1 were confirmed by immunoblot (Figure 3C), while no CD9 expression was detected. EVs were stained with CFDA-SE to discriminate intact vesicles from debris and membrane fragments (Figure 3D). Using a mixture of fluorescent dimensional beads, specific size gates were considered (≤100 nm, from 100 to 160 nm and from 160 to 900 nm) (Figure 3E, left panel). The percentage of CFDA-SE events falling in the three gates showed that P3 regularly fed hBM-MSC-derived EVs were mostly in the gates ≤100 nm and from 100 nm to 160 nm, confirming the average size previously observed by NTA. The expressions of CD81 and CD63 were also validated by flow cytometry (Figure 3E, right panels), as well as CD9’s low expression.

### 3.3. Diabetic Fractured Mouse Model Shows Increased Inflammation and Reduced Mineral Matrix Content

To validate our main hypothesis in vivo, we adopted a well-established preclinical model of delayed fracture healing [21,32] using the experimental plan depicted in Figure 4A. Diabetic induction with STZ was successful, resulting in a mean of plasma glucose level of 23.9 ± 5.3 mM in comparison to control animals (injected with HBSS) having 11.2 ± 0.57 mM (*p* ≤ 0.05) (Figure 4B). Diabetic mice also showed a reduced but steady weight throughout the study, 26.34 ± 1.17 g for STZ-treated animals and 32.3 ± 1.92 g (*p* ≤ 0.05) for control animals (Figure 4C). X-ray scans taken immediately after the creation of the mice’s femur transverse fractures (Figure 4D) and those taken two days later (Figure 4E) directed the procedure for injecting cells into the fracture site. Additionally, X-rays were used to track the healing of fractures at days seven (Figure 4F) and 23 (Figure 4G). The results indicated that diabetes had a negative effect on bone regeneration only at the earlier time-point (seven days). All animals treated with STZ had reduced mineral content at the fracture site compared to untreated animals at this time-point, as seen from the μCT-analyzed parameters bone volume/total volume (BV/TV(%)) (*p* ≤ 0.05), connectivity (number of trabeculae) (*p* ≤ 0.05), connectivity density (number of trabeculae per unit volume, Conn.D (mm^−3^)) (*p* ≤ 0.05), bone surface (BS(mm^2^)) (*p* ≤ 0.05), trabecular thickness mean (Tb. Th. Mean (mm)) (*p* ≤ 0.05), trabecular thickness max (Tb. Th. Max (mm)) (*p* ≤ 0.05) and trabecular spacing max (Tb. Sp. Max (mm)) (*p* ≤ 0.05), but not for the trabecular spacing mean (Tb. Sp. Mean (mm)) (Figure 4H). No differences were seen in the mineral content in any μCT measurement assessed between diabetic and non-diabetic mice at the longer time-point (23 days) (Figure 4I).

Blood samples collected on day two (four hours after treatment) and seven days following fracture from STZ-treated and untreated animals were evaluated to establish the inflammatory fingerprint of the animal model as a result of the establishment of the diabetic condition. Data measuring 23 inflammation-related biomarkers indicated that pro-inflammatory mediators such as IL-1α, IL-6, IL-9, IL-12(p40), IL-12(p70), IL-17, eotaxin, G-CSF, IFN-γ, KC, MCP-1, MIP-1β and RANTES were significantly different in fractured diabetic compared to fractured non-diabetic animals at two days post fracture, (*p* ≤ 0.05) (Figure 4I). No significant difference was found for IL-1β, IL-2, IL-3, IL-4, IL-5, IL-10, IL-13, GM-CSF, MIP-1α and TNF-α in STZ-induced diabetic fractured mice compared to normo-glycemic fractured mice (Figure 4J,K). 

At day seven IL-1α, IL-1β, IL-9, IL-12(p40), IL-12(p70), IL-13, IL-17, eotaxin, G-CSF, IFN-γ, KC, MCP-1 and RANTES were significantly different in fractured diabetic (STZ-treated) compared to fractured non-diabetic (STZ-untreated) animals (*p* ≤ 0.05) (Figure 4I,J). No significant differences were seen for IL-2, IL-3, IL-4, IL-5, IL-6, IL-10, GM-CSF, MIP-1α, MIP-1β and TNF-α in STZ-induced diabetic fractured mice over control non-diabetic fractured mice (Figure 4J,K).

In addition, IL-1α, IL-4, IL-6, IL-9, IL-12(p70), IL-17, G-CSF, GM-CSF, IFN-γ, KC and MIP-1β levels were significantly higher (*p* ≤ 0.05) at day two compared to day seven, whereas IL-2, IL-12(p40), eotaxin and RANTES levels were significantly lower (*p* ≤ 0.05) (Figure 4L).

### 3.4. Starved Cells Enhance Anti-Inflammatory Effects and Favor Earlier Soft Callus Formation and Mineral Matrix Deposition

Blood samples taken from the animals two days after fracture (four hours after cell and EVs injections) revealed similar levels for starved and regularly fed cells for IL-3, IL-4, IL-5, IL-9, IL-12(p70), GM-CSF, IFN-γ, KC, MCP-1, MIP-1α, RANTES and TNF-α levels; however, their levels in starved and regularly fed cells were lower than in mice treated with EVs (*p* ≤ 0.05 * with respect to EVs, + with respect to reg fed cells,) (Figure 5A–C). At the day two time-point, only IL-17 was greater in the starved cells than in both the EVs and regularly fed cells (*p* ≤ 0.05), whereas IL-1α, IL-1β, IL-2, IL-10 and IL-13 were lower in the starved cell mice than in animals injected with the other two treatments (*p* ≤ 0.05 * with respect to EVs, + with respect to reg fed cells) (Figure 5A–C).

At seven days, although several cytokines were significantly lower in the starved cell-injected animals than in the regularly fed or EV-injected animals ((IL-1α, IL-12(p70), IL-17, GM-CSF, IFN-γ, MIP-1α, MIP-1β and RANTES, *p* ≤ 0.05 * with respect to EVs, + with respect to reg fed cells)), the expression of IL-6, IL-13, eotaxin and KC were higher after the injection of the starved cells, compared to the animals treated with EVs and regularly fed cells (Figure 5D–F). When the data from the two time-points were compared between animals injected with regularly fed cells, it was seen that IL-1α, IL-6, IL-9, IL-10, IL-12(p70), IL-17, G-CSF, KC, MCP-1, MIP-1α and RANTES reached their secretion peak at the seventh day time-point with IL-2, IL-3, IL-13 and TNF-α peaking at the second day time-point (* *p* ≤ 0.05) (Figure 5G,H).

Comparison between the seventh and second day time-points in animals treated with EVs demonstrated that IL-1α, IL-1β, IL-10, IL-12(p70), IL-17, eotaxin, GM-CSF, KC, MCP-1 and MIP-1α were higher at the seventh day time-point than at the second day time-point. IL-3, IL-4, IL-6, IL-9, IL-12(p70), IL-13, IFN-γ, MIP-1β, RANTES and TNF-α levels were higher at the early time-point (* *p* ≤ 0.05) (Figure 5G,I).

Finally, starved cell-treated animals had higher levels of IL-13 and KC at the seventh day time-point compared to IL-1α, IL-4, IL-12(p70), IL-17, IFN-γ, MIP-1α and RANTES at the second day time-point (* *p* ≤ 0.05) (Figure 5G,J).

To evaluate quantitative variations in the microarchitecture of a newly formed mineral matrix, µCT analyses were performed on mouse femurs injected with hBM-MSC, EVs, and diabetic fractured and non-diabetic fractured controls at two different time-points. After seven days, there appears to be denser cortical lightly mineralized material adjacent to the fracture site highlighting the soft callus presence as seen in the 3D reconstructions (blue) (Figure 6A) and in Figure 6B–E with decreasing levels (blue) starting from starved (Figure 6C) to EVs (Figure 6D) and regularly fed (Figure 6E) as compared to diabetic fracture controls (Figure 6B). Quantitative analysis of μCT sequential images confirmed that starved hBM-MSC were able to promote greater mineral matrix deposition than EVs and regularly fed cells on the seventh day post fracture, (*p* ≤ 0.05, * with respect to EVs, + with respect to reg fed cells) (Figure 6F,G).

Indeed, the starved cells’ injection resulted in a greater BV/TV ratio (1.46 ± 0.19), connectivity (3.41 ± 0.79), Conn.D (mm^−3^) (4.23 ± 1.6) (*p* ≤ 0.05 * with respect to EVs, + with respect to reg fed cells) compared to regularly fed cells and EV-injected animals and BS (mm^2^) (1.85 ± 0.5) (*p* ≤ 0.05 * with respect to EVs) compared to EV-treated cells (Figure 6F,G). No significant difference was seen between the treatments for trabecular Tb. Th. Mean (mm), Tb. Th. Max (mm), Tb. Sp. Mean (mm) and Tb. Sp. Max (mm) (Figure 6F,G). With respect to the diabetic fractured mice, preconditioned/starved cells induced a significant improvement in BV/TV, connectivity, Conn.D (mm^−3^) and BS (mm^2^) (*p* ≤ 0.05 ** with respect to diabetic fractured) (Appendix A). No significant difference was seen between the treatments for trabecular Tb. Th. Mean (mm), Tb. Th. Max (mm), Tb. Sp. Mean (mm) and Tb. Sp. Max (Appendix A).

At the 23 day time-point, a substantial amount of cortical and trabecular newly formed mineralized matrix (grey) and the presence of a lightly mineralized soft callus (blue) were evidenced as seen in Figure 7A–D; however, no significant difference in any of the investigated parameters was seen across all the tested treatments (Figure 7E,F and the diabetic fracture control (Appendix A).

## 4. Discussion

When a fracture occurs, it impacts not only the bone tissue but also the nearby soft tissues, the local vasculature and the bone marrow. As a result, a hematoma develops, which is characterized by a limited supply of oxygen and nutrients and acute inflammation in the early stages [38]. Neutrophils are among the first inflammatory cells to colonize the fracture site, generating pro-inflammatory and chemotactic mediators that drive the recruitment of a large group of inflammatory cells, mainly monocytes and macrophages [39]. Fracture hematoma and inflammatory lesions often resolve within a few days to a week following the fracture, and are replaced at the soft callus stage by developing neo-vasculature and granulation tissue rich in mesenchymal cells [40]. Subsequently, chondroblasts develop the soft callus via cartilage apposition two to four weeks after the injury, providing quick structural support to the fracture. Finally, the soft callus serves as a scaffold for endochondral bone formation [40] and osteoprogenitor cell differentiation, followed by the deposition of woven bone on the cartilage leading to the hard callus phase months after the initial injury [40].

Inflammatory cells produce a variety of cytokines, influencing cells in both positive and negative ways and the various phases of bone regeneration [2,25].

In fact, neutrophils release pro-inflammatory and chemotactic mediators such as IL-6 and MCP-1 (CCL2) [25] during the initial step. Monocyte/macrophages secrete many different cytokines as well, such as interleukin-1α and β (IL-1α and β), which suppress matrix mineralization and osteoblastic differentiation [26]. TNF-α reduces the chondroblasts’ viability [27], inhibits osteogenic differentiation [26] and reduces bone formation in mice [28]. IL-4 and interferon γ (IFN-γ) are also significantly anti-osteogenic [29]. Moreover, IFN-γ and TNF-α synergistically induce BM-MSC apoptosis and significantly inhibit bone formation in vivo [28], while IL-17 stimulates the nuclear factor kappa B (NF-κB) signaling pathway and impairs the differentiation of BM-MSC [2].

At later stages many different studies have demonstrated that inflammatory and chemotactic mediators, such as TNF-α, IL-4, IL-6, IL-13, IFN-γ, KC and MCP-1 released from macrophages, lymphocytes and eosinophils, also stimulate the recruitment of fibroblasts, mesenchymal stem cells (MSC) and osteoprogenitor cells from their niches [25,41,42,43]. For example, it was seen that mice missing the TNF-α receptor gene have a significant delay in the chondrocyte differentiation [42] and endochondral ossification [44]. Interleukin 6 (IL6) has been similarly implicated in bone healing delay as well as the lack of reduced fracture healing [32].

Chronic hyper-inflammatory conditions, such as those triggered by diabetes [1,21,22,23], de facto impair even more fracture healing by increasing neutrophils [1], monocytes and dendritic cells [24], decreasing soft callus formation and enhancing osteoclastic activity and bone resorption [30], thereby delaying fracture healing [1,22,31]. Furthermore, changes in neutrophil and monocyte/macrophage populations produce pro-inflammatory and chemotactic mediators such as IL-6, MCP-1 (CCL2), IL-1, TNF-α, IL-4 and IFN-γ, all of which have anti-osteogenic activity, also leading to fracture healing impairment [25,26,27,28,29].

Such a hyper-inflammatory milieu was recapitulated in the diabetic STZ-induced animal models used in this research as well as by McCauley J. et al. [32].

MSC have long been proposed as a biological treatment in immunotherapy and regenerative medicine due to their multilineage differentiation capacity and immunological regulation. The cellular environment surrounding MSC or the pathological conditions, however, can negatively regulate or restrict the potential of the MSC, as in the case of fractures in diabetic conditions. This environment leads to a sharp decline in the injected cell viability [10] or a reduced activity of the resident MSC as a result of the activation of apoptotic [7] signaling brought on by the depleted nutrient and oxygen status as well as the hyper-inflammatory state. It is also true that an MSC’s behavior is influenced by and adapts [10,17] to external stimuli; therefore, the success of mesenchymal stem/stromal cells as immunotherapeutic agents or in regenerative medicine may depend on understanding how to activate or inhibit their adaptive processes before or during therapeutic scenarios.

In our study, we subjected our cells to an intense preconditioning procedure, which, as demonstrated in our previously published paper, results in a significant reduction in the number of cells during starvation and may increase the presence of waste/toxic molecules such as lactate and ammonia [45]. This likely contributed to increasing the osmolarity and ionic strength of the culture medium [17]. Although we did not measure the level of ammonia in our previous research, we found that lactate transiently increased in the culture medium until day 27 before decreasing towards the end of the starving period [17]. Recent research demonstrated that lactate supplementation can decrease both lactate and ammonia levels in culture [46]. These results, along with the presence of albumin, both as messenger and protein, in starvation-induced conditions [17], which is known to be a lipid and metal ion transporter with antioxidant and buffering properties, imply that the starvation process activates specific cellular adaptive mechanisms that lessen the negative effects.

CD146 reduction is not a surprising event that was reported to be dependent on the culture conditions and aging leading to reduced osteoblastic activity [47]. The fact that our culture medium is composed of α-MEM, which was not seen to induce CD-146 reduction, even at higher passages (P8) [47], suggests that the starvation process is the cause of its decrease and, as a consequence, a reasonable reduction in the hBM-MSC osteoblastic potential. On the contrary, a comparison of the results obtained from P6 starved cells and P3 regularly fed cells showed no reduction in their osteoblastic capability.

In addition, the results showed that preconditioned/starved cells possibly adapt to the in vitro harsh conditions by secreting higher amounts of growth factors, such as IGFBP2, IGFBP3, IGFBP6, HGF, TGFβ2, PDGFα, VEGFR2 and IGF2) known as critical mediators in the process of fracture healing [2,25,43,48,49,50,51,52,53,54], than cells grown in a fresh medium (Figure 8) [17]. Therefore, it could be speculated that this may also occur in vivo promoting neovascularization (VEGFR2, IGFBPs) [49,50,51,52], the formation of granulation tissue, and the recruitment and differentiation of autologous osteoblastic progenitor cells (IGFBP2, IGF2 TGFβ2, HGF [2,53,54] and MSC (TGFβ2, PDGFα) [2] in the context of fracture healing.

In addition, the analysis of the pro-inflammatory cytokines and chemokines revealed that, compared to EV-treated animals, starved cell- and regularly fed cell-injected animals had a negligible influence on pro-inflammatory cytokine synthesis at both time-points (such as IL-1α, IL-1β, TNFα and IFN-γ), thus corroborating a putative inhibitory effect for these cells on pro-inflammatory cytokine production, which has been widely and previously described for MSC [3,55].

Nonetheless, the highest levels of IL-17 and IL-6 at the earlier time-point, as well as IL-13, KC and eotaxin at the seventh day time-point, suggest that starving cells outperform both regularly fed cells and EVs (Figure 8). Indeed, recent findings suggest that an increased and early presence of IL-17-producing γδ T cells promotes the osteoblastic differentiation of progenitor cells [56] via two temporally defined mechanisms: during the early inflammatory cascade via C/EBPβ, independent of the canonical Wnt pathway and β-catenin signaling; later, during the reparative/remodeling phase, IL-17 acts via C/EBPδ [57], which could be consistent with the observations of increased IL-17 plasma concentrations in mice treated with starved cells.

It is widely recognized that macrophages can play a dual role in fracture healing, being either pro-inflammatory during the initial phases as a result of the classical activation pathway, via IFN-γ, or anti-inflammatory through the alternative activation pathway, via IL-4 and IL-13 [2,43]. In fact, the alternative activation pathway promoting macrophage activity, collagen deposition and tissue homeostasis has been demonstrated to favor fracture healing [48]. Therefore, here, it may be suggested that the induced secretion of IL-13, by starved cells, or IL-4, by EVs, plays a role in alternative macrophage polarization favoring tissue homeostasis and matrix secretion [32,48], accelerating the development of a soft callus, and thus contributing to early fracture stabilization. Local vascular regeneration and infiltration within the fracture callus are just as important as inflammation resolution or bone regeneration. KC is a chemokine that is released by dendritic cells and its presence at the fracture site early during the healing process promotes neutrophil recruitment [32] and vascular regeneration by enhancing monocyte adhesion and density around the developing arterial collaterals, leading to successful fracture healing [43,49]. The fact that KC increases following the injection of starved cells implies that its availability in animals on day seven post fracture supports its beneficial effect on cell recruitment and revascularization. Furthermore, the increased osteoclast recruitment and resorption of bone fragments at the fracture site is attributed to eotaxin, which acts as a natural antagonist for the C-C motif chemokine receptor 2 and an agonist for CCR5 [32,58]; therefore, the increased eotaxin presence, favoring osteoclast recruitment and resorption of the bone fragments during the initial stages of fracture healing, positively influences bone regeneration (Figure 8).

Such positive effects could suggest that starved cells, in addition to inducing a quicker soft callus formation, which is per se already interesting because of the evidence of a delayed soft callus formation in diabetic conditions [30], also favor the faster deposition of mineralized material that translates into an improved and earlier fracture stabilization at seven days post fracture. In fact, Kayal et al. demonstrated that under diabetic conditions [34,38], the delayed fracture healing is related to an increased chondrocytes’ apoptosis, via the TNF-α/FOXO1 axis, and a premature cartilage resorption due to an increased osteoclastic activity [27,30]. These results support previous findings, revealing that the starvation approach had the most favorable influence on growth factor release, mineral matrix deposition and anti-inflammatory activity over EVs and regularly fed cells derived from the same donor. Further studies based on histological analysis will be needed to fully understand the response to the proposed cell therapy. However, on the downside, such favorable benefits appear to be limited to the early phases of fracture healing, as the fraction of mineralized matrix in normo-glycemic, diabetic and diabetic-treated animals was comparable on day 23, which is consistent with previous findings during fracture healing [21,30] (Figure 8).

## 5. Conclusions

This study demonstrated that the long-term starvation of hBM-MSC enhances the secretion of trophic factors, improving the early stage of fracture healing in diabetic mice by decreasing pro-inflammatory cytokine expression and increasing anti-inflammatory cytokine production, angiogenesis and progenitor cell recruitment, leading to an earlier soft callus formation, mineralization and stabilization of the fracture. Our findings are consistent with those by Zhang J. et al. [5], who demonstrated that hypoxia preconditioning improves fracture healing and suggest, as confirmed by Moya A. et al. [10], that starvation increases cell survival, indicating that preconditioning studies are needed to improve the regenerative and anti-inflammatory effects of grafted cell survival after transplantation.

This work supports the idea that the in vitro crosstalk between the cells and stimuli is an important factor that has yet to be deeply studied to fulfill MSC’s therapeutic potential, thus pointing out that in vitro preconditioning of stem/stromal cells may be crucial for refining and defining new criteria for future cell therapy applications. That said, a better understanding of stem/stromal cell adaption and survival processes may lead to increased cell viability and, thus, an improvement in therapeutic efficacy after transplantation.

Since the main concern about using stem cells in clinical settings is the need for cell expansion outside their quiescent niches, we previously suggested in our published research [17] that the MSC culture in a xenogen-free medium would be the preferred choice for future MSC in vitro amplification due to its reduced nutritional content. In addition, we suggest that deep studies of the starvation approach at different time-points, both in normoxia or hypoxia, and for different periods, will also reveal the survival, regenerative and anti-inflammatory properties of the cells. Importantly, the therapeutic potential of the MSC also has to be explored by testing the properties of the EVs retrieved from starved cells.

In summary, there is still more that can be done to fully comprehend and appreciate the preconditioned MSC’s and EV’s whole potential, and this may also be revealed by pushing them to their limit, even though this is not strictly necessary.

## Figures and Tables

**Figure 1 cells-11-02696-f001:**
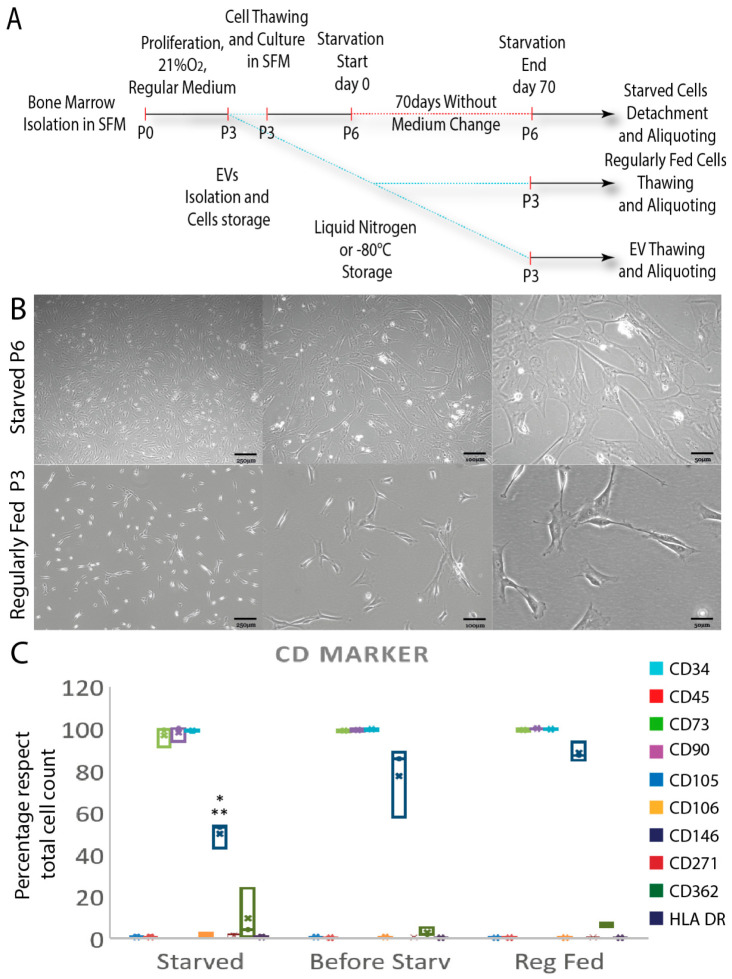
Starved hBM-MSC changed morphologically but retained their phenotype. (**A**) Diagram depicting the in vitro starving procedure and time-points observed throughout the investigation. (**B**) Progressive phase contrast images of 70-day-starved hBM-MSC show fibroblast-like cells with a uniform appearance with minimal differences between P6 starving cells and P3 regularly fed cells. (**C**) Starved hBM-MSC retained surface marker expression by flow cytometry analysis; only CD 146 decreased significantly post starvation with respect to regularly fed and the same cells before starvation. Scale bars: 250–100–50 μm. Results are expressed as percentage ± SD with respect to total events count of three independent experiments with *, ** *p* ≤ 0.05 indicating significance (one-way ANOVA followed by Bonferroni’s post hoc test). Abbreviations: human bone marrow mesenchymal stem/stromal cells (hBM-MSC); extracellular vesicles (EVs); serum-free; medium (SFM); cluster of differentiation (CD).

**Figure 2 cells-11-02696-f002:**
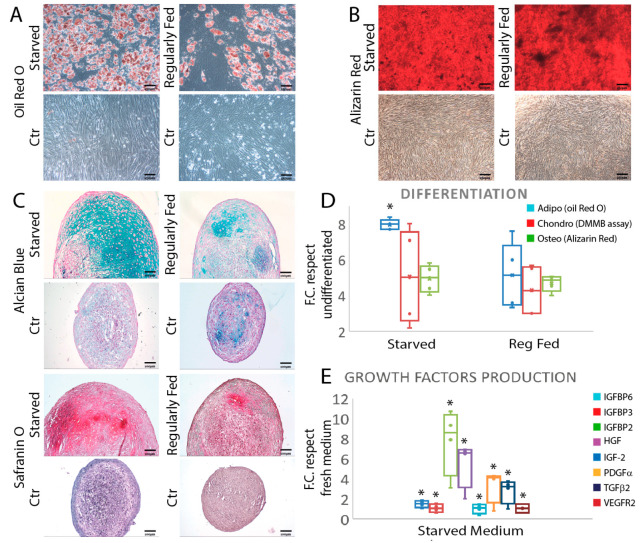
Starved cells retain their ability to differentiate and survive secreting trophic factors: (**A**) Oil red O staining comparing the hBM-MSC adipocytic differentiation potential in regularly fed and starved cell populations. (**B**) Alizarin red staining showing the osteoblastic potential of the tested cells in the same groups of A. (**C**) Alcian blue and safranin O staining evidencing the chondroblastic potential of the starved and regularly fed cells. (**D**) Whisker plot comparing the differentiation potential of the tested hBM-MSC. Results are normalized with respect to control cells cultured in proliferation medium. Results are presented as mean ± SD of three independent experiments. * *p* ≤ 0.05 (one-way ANOVA followed by Bonferroni’s post hoc test). (**E**) Starvation led to increased growth factor secretion by hBM-MSC. Results are presented as mean ± SD of two independent experiments. * *p* ≤ 0.05 (one-way ANOVA followed by Bonferroni’s post hoc test). Abbreviations: control (Ctr); fold change (F.C.); dimethyl-methylene blue assay (DMMB); insulin growth factor binding protein 6 (IGFBP6, 3, 2); hepatocyte growth factor (HGF); insulin growth factor 2 (IGF2); platelet-derived growth factor (PDGFα); transforming growth factor 2 (TGFβ2); vascular endothelial growth factor receptor 2 (VEGFR2).

**Figure 3 cells-11-02696-f003:**
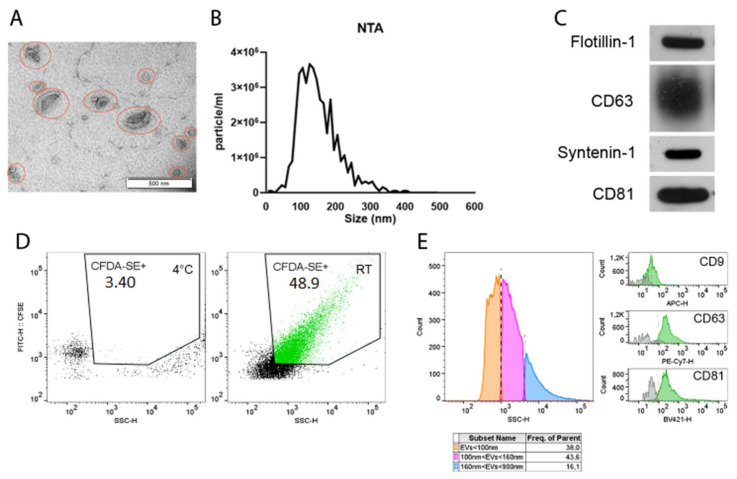
Extracellular vesicles’ characterization: (**A**) TEM micrographs of P3 regularly fed hBM-MSC-derived EVs (red circles). (**B**) Nanoparticle tracking analysis showing the size distribution of P3 regularly fed hBM-MSC-derived EVs. (**C**) Immunoblot analysis of P3 regularly fed hBM-MSC-derived EVs. Specific expressions of CD63, CD81, flotillin-1 and syntenin-1 were investigated. (**D**) CFDA-SE staining by non-conventional flow cytometry. Left panel shows the 4 °C control. Right panel shows CFDA-SE staining at RT. Green areas identify CFDA-SE positive events. (**E**) Size distribution of EV subtypes. Three-dimensional gates were considered: EVs ≤ 100 nm (orange), 100 nm ≤ EVs ≤ 160 nm (pink) and 160 nm ≤ EVs ≤ 900 nm (blue). Right panels show CD9, CD63 and CD81 positive events falling within the CFDA-SE gate. Data are representative of at least three independent experiments. Abbreviations: nanoparticle tracking analysis (NTA); human bone marrow mesenchymal stem/stromal cells (hBM-MSC); extracellular vesicles (EVs); transmission electron microscope (TEM); nanoparticle tracking analysis (NTA); cluster of differentiation (CD); carboxyfluorescein diacetate succinimidyl ester (CFDA-SE).

**Figure 4 cells-11-02696-f004:**
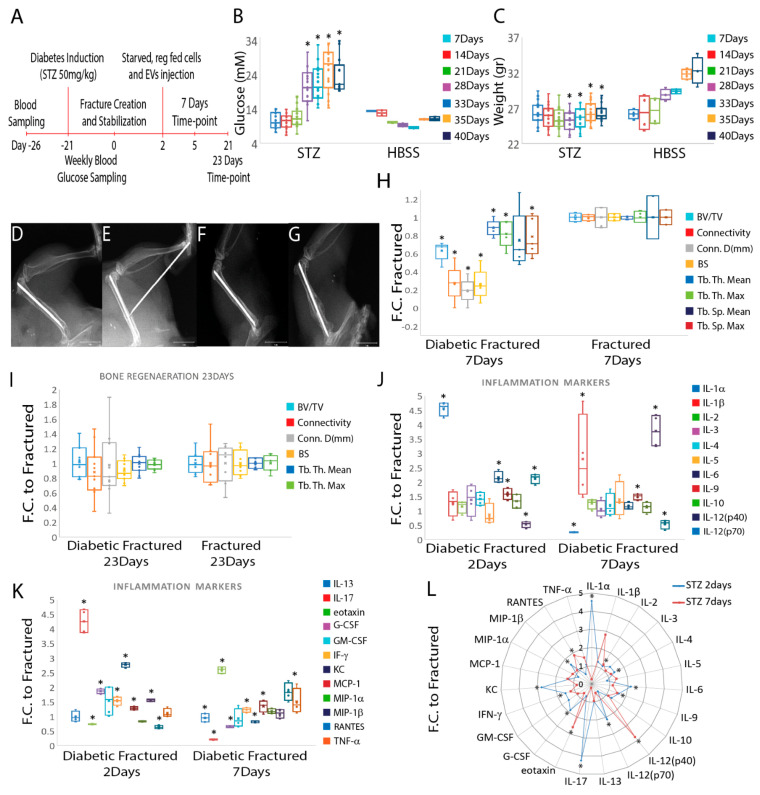
In vivo approach summary and comparison of regenerative and inflammatory cytokine expression in diabetic and non-diabetic mice at two and seven days post fracture. (**A**) In vivo experimental plan employed in this study. (**B**) Diabetes was successfully induced with STZ compared to control mice. The data are presented as mean ± SD. * *p* ≤ 0.05 (sample size four to sixteen animals/group, Student’s *t*-test and paired *t*-test). (**C**) The bodyweight of diabetic mice was consistently lower than that of control animals. The data are shown as mean ± SD. * *p* ≤ 0.05 (sample size four to sixteen animals/group, Student’s *t*-test and paired *t*-test). (**D**–**G**) X-ray scans of the mouse femur immediately after fracture (**D**), after two days (**E**), which directed the approach for treatment into the fracture site, and after seven (**F**) and 23 days (**G**). (**H**) µCT quantitative analysis of diabetic mice fracture revealed a decreased mineral matrix content when compared to untreated animals at the seventh day time-point. The data are normalized to non-diabetic fractured animals and are presented as mean fold change ± SD. *p* ≤ 0.05 * respect to EVs, +Cells. (**I**) No difference was seen between the diabetic and non-diabetic animals at the 23rd day time-point for any one of the analyzed μCT parameters. The data are normalized to non-diabetic fractured animals and are presented as mean fold change ± SD (sample size four to six animals/group, one-way ANOVA followed by Fisher’s post hoc test). (**J**–**K**) Comparison of inflammatory cytokine expression in diabetic and non-diabetic mice at two and seven days post fracture with respect to fractured animals. (**L**) Comparison of inflammatory cytokine expression in diabetic fractured mice at two and seven days. The data are normalized to non-diabetic fractured animals and are presented as mean fold change ± SD. * *p* ≤ 0.05 (sample size four animals/group, Student’s *t*-test and paired *t*-test). Scale bar: 1 mm. Abbreviations: streptozotocin (STZ); extracellular vesicles (EVs); fold change (F.C.); bone volume/total volume, (BV/TV); connectivity density, (Conn.D (mm^−3^)); bone surface, (BS(mm^2^)); trabecular thickness mean, (Tb. Th. Mean (mm)); trabecular thickness max, (Tb. Th. Max (mm)); trabecular spacing mean, (Tb. Sp. Mean (mm)); trabecular spacing max, (Tb. Sp. Max (mm)).

**Figure 5 cells-11-02696-f005:**
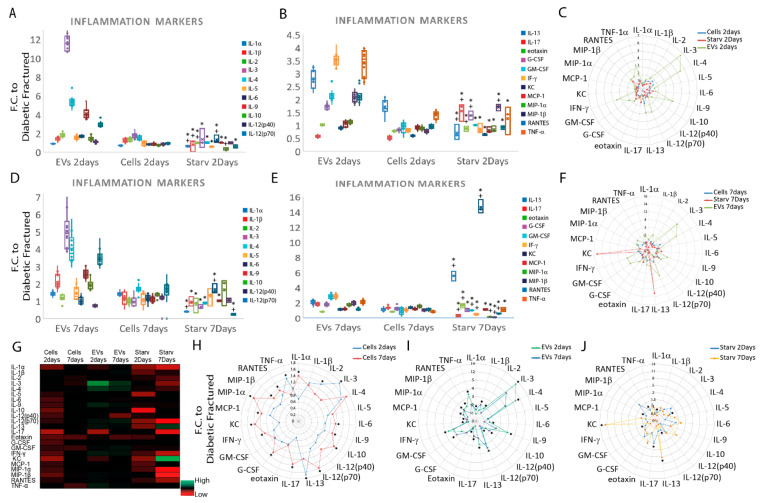
Comparison of inflammatory cytokine expression in diabetic mice treated with starved, regularly fed cells and EVs at two and seven days post fracture. (**A**,**B**) Graphs comparing the inflammatory cytokines in plasma at the day two time-point (sample size three to eight animals/group, one-way ANOVA followed by Tukey’s post hoc test). (**C**) Radar graph highlighting the differences between the different treatments at day two (sample size three to eight animals/group, paired *t*-test). (**D**,**E**) Graphs comparing the inflammatory cytokines at day seven (sample size three to eight animals/group, one-way ANOVA followed by Tukey’s post hoc test). (**F**) Radar graph highlighting the differences between the different treatments at the day seven time-point (sample size three to eight animals/group, paired *t*-test). (**G**) Heat map showing direct comparison of the inflammatory cytokines within the different treatments and the two time-points. (**H**–**J**) Radar graph highlights the differences between animals treated with cells (**H**), EVs (**I**), starved cells (**J**) at the different time-points. The data are normalized to fractured diabetic animals who had only PBS injection and presented as mean fold change ± SD. *p* ≤ 0.05 * respect EVs, + cells. Abbreviations: regularly fed cells (cells); starved cells (starved); extracellular vesicles (EVs); fold change (F.C.); interleukin (IL-1α, 1β, 2, 3, 4, 5, 6, 9, 10, 12(p40), 12(p70), 13, 17); granulocyte colony-stimulating factor (G-CSF); granulocyte-macrophage colony-stimulating factor (GM-CSF); interferon-γ (IFN-γ); keratinocytes-derived chemokine (KC); monocyte chemotactic protein 1 (MCP-1); macrophage inflammatory protein-1α, 1β (MIP-1α and MIP-1β); regulated on activation, normal T cell expressed and secreted (RANTES); tumor necrosis factor-α (TNF-α).

**Figure 6 cells-11-02696-f006:**
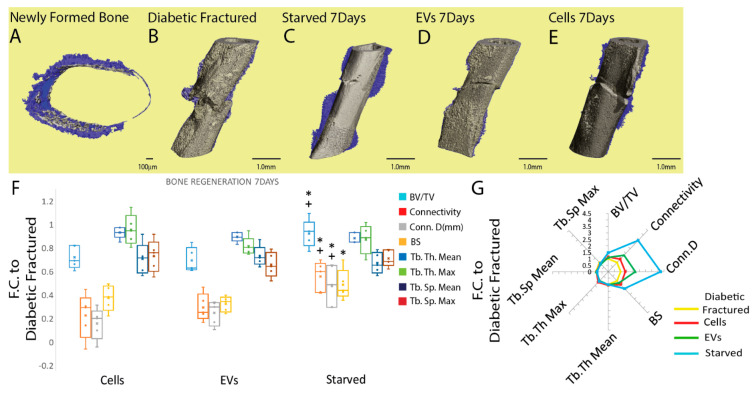
Analysis of the newly formed mineralized tissue five days after cells (starved and regularly fed) and EVs injections at the fracture site. (**A**) Three-dimensional fracture site reconstruction showing the presence of denser cortical lightly mineralized material adjacent to the fracture site highlighting the soft callus presence (blue) seven days post fracture. (**B**–**E**) Three-dimensional reconstruction of µCT images from diabetic mice and diabetic mice treated with starved cells, EVs and regularly fed cells produced from the same donors showing the presence of decreasing amounts of denser lightly mineralized matrix (blue), delineating the soft callus formation, from (**B**–**D**). (**F**,**G**) Starved cell administration had a positive effect on mineral matrix deposition at the earlier time-point (7 days) for BV/TV ratio, connectivity, Conn.D (mm^−3^) compared to regularly fed cell- and EV-injected animals and BS (mm^2^) compared to EV-treated cells. No significant difference was seen between the treatments for trabecular Tb. Th. Mean (mm), Tb. Th. Max (mm), Tb. Sp. Mean (mm) and Tb. Sp. Max (mm). Data were normalized to diabetic animals with femoral fracture but receiving PBS injection. Results are presented as mean fold change ± SD. *p* ≤ 0.05 * with respect to EVs, +cells (sample size four to five animals/group, one-way ANOVA followed by Fisher’s post hoc test). Scale bars: 100 µm, 1 mm. Abbreviations: micro-computed tomography (µCT); regularly fed cells (cells); starved cells (starved); extracellular vesicles (EVs); fold change (F.C.); bone volume/total volume (BV/TV); connectivity density (Conn.D (mm^−3^)); bone surface (BS(mm^2^)); trabecular thickness mean (Tb. Th. Mean (mm)); trabecular thickness max (Tb. Th. Max (mm)); trabecular spacing mean (Tb. Sp. Mean (mm)); trabecular spacing max (Tb. Sp. Max (mm)).

**Figure 7 cells-11-02696-f007:**
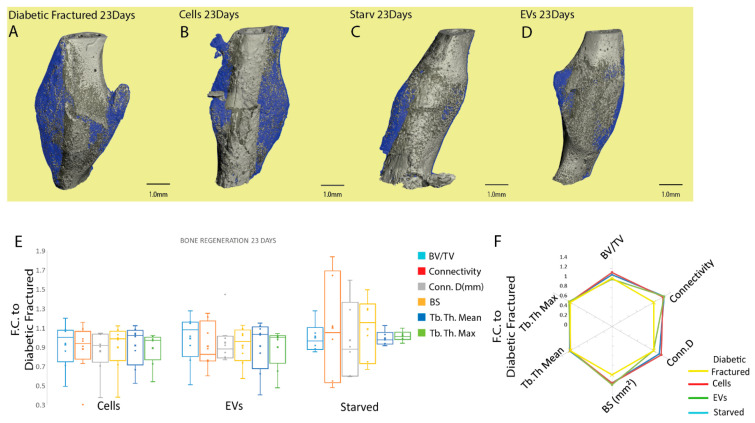
Analysis of the newly formed mineralized tissue at 21 days after cell and EV injection at the fracture site. (**A**–**D**) Three-dimensional femur reconstruction showing cortical and trabecular newly formed mineralized matrix (grey) and the presence of a lightly mineralized soft callus (blue) at the 23 day post-fracture time-point. (**E**,**F**) Treatment with regularly fed, starved cells and EVs had no effect on bone regeneration at the later time-point (23 days) as evident by µCT parameters BV/TV ratio, connectivity, Conn.D (mm^−3^), BS (mm^2^), trabecular Tb. Th. Mean (mm), Tb. Th. Max (mm). Data are normalized to diabetic animals that had a femoral fracture but only received a PBS injection. The results are presented as mean fold change ± SD (sample size five to eight animals/group, one-way ANOVA followed by Fisher’s post hoc test). Scale bars: 1 mm. Abbreviations: micro-computed tomography (µCT); regularly fed cells (cells); starved cells (starved); extracellular vesicles (EVs); fold change (F.C.); bone volume/total volume (BV/TV); connectivity density (Conn.D (mm^−3^)); bone surface (BS(mm^2^)); trabecular thickness mean (Tb. Th. Mean (mm)); (*p* ≤ 0.05); trabecular thickness max (Tb. Th. Max (mm)); trabecular spacing mean (Tb. Sp. Mean (mm)); trabecular spacing max (Tb. Sp. Max (mm)).

**Figure 8 cells-11-02696-f008:**
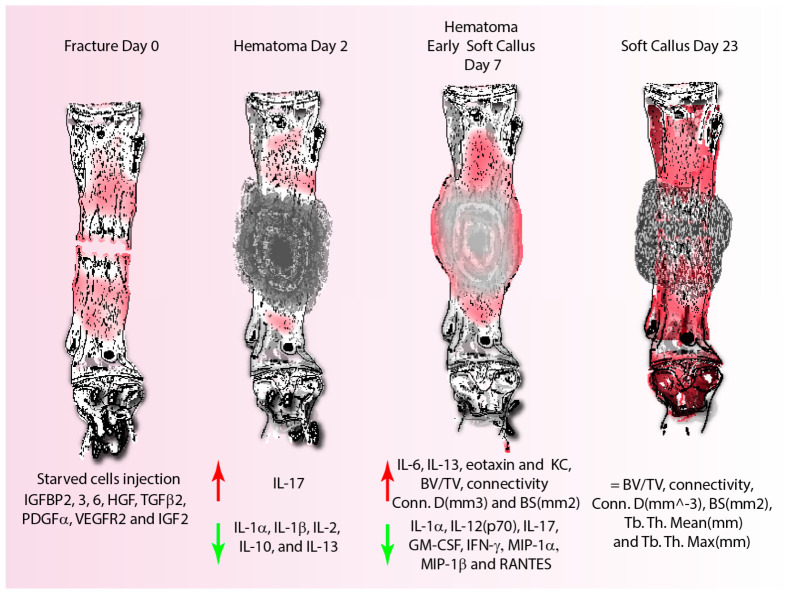
Graphic outlining the effects of hBM-MSC-starved cell injection on bone healing, as well as the released cytokines and growth factors involved in the early phases of fracture healing. Results demonstrate that the starvation process induces the production of different growth factors such as IGFBP2, 3, 6, HGF, TGFβ2, PDGFα, VEGFR2 and IGF2, which induce the secretion of cytokines such as IL-17 and repress the production of IL-1α, β, 2, 10 and 13 at the earliest time-point. At day 7 after fracture, starved cells favor the expression of IL-6, 13 and KC, increase the presence of mineralized matrix and, presumably, the soft callus formation at the fracture site, with a decrease in the expressions of IL-1α, 12(p70), 17, GM-CSF, IFN-γ, MIP-1α, β and RANTES. However, at 23 days post fracture, no difference was seen within the tested conditions. Abbreviations: streptozotocin (STZ); bone volume/total volume (BV/TV); connectivity density (Conn.D (mm^−3^)); bone surface (BS(mm^2^)); trabecular thickness mean (Tb. Th. Mean (mm)); trabecular thickness max (Tb. Th. Max (mm)); trabecular spacing mean (Tb. Sp. Mean (mm)); trabecular spacing max (Tb. Sp. Max (mm)); interleukin (IL-1α, 1β, 2,6, 10, 12(p70), 13, 17); granulocyte-macrophage colony-stimulating factor (GM-CSF); interferon-γ(IFN-γ); keratinocytes-derived chemokine (KC); macrophage inflammatory protein-1α, 1β (MIP-1α and MIP-1β); regulated on activation, normal T cell expressed and secreted (RANTES); insulin growth factor binding protein 6 (IGFBP6, 3, 2); hepatocyte growth factor (HGF); insulin growth factor 2 (IGF2); platelet-derived growth factor (PDGFα); transforming growth factor 2 (TGFβ2); vascular endothelial growth factor receptor 2 (VEGFR2).

## Data Availability

Data are available from the corresponding author upon request. In data are available from the corresponding author upon request.

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
