# Peer review of "Regenerative and Anti-Inflammatory Potential of Regularly Fed, Starved Cells and Extracellular Vesicles In Vivo"

_cells, 2022, doi:10.3390/cells11172696_

Round 1

Reviewer 1 Report

The authors present a very complete study on the possible interest of pre-conditioning mesenchymal stem cells in stringent conditions prior to implantation. The authors hypothesis that promoting cell quiescence would potentially results in better cellular adaptation upon in vivo implantation in the context of bone fracture healing, with emphasis on their anti-inflammatory and regenerative properties.  

In the first part of their study, the authors focused on the in-vitro characterization of starved MSCs compared to classically cultured MSC in serum free medium (called regular MSC in this report). Deep analysis of surfaces markers, and cytokine production were performed, as well as tri-lineage differentiation capacity of the starved cells versus regular cells, or versus extra-cellular vesicles isolated from regular-cells conditioned media.

In the 2nd part of this work, the authors developed/used a diabetic fracture repair model in mice (C57BL/6J) for the study the in-vivo efficiency of cells / conditioned media prepared as described in the first part on bone healing and repair.     

 In general, this is a nicely designed and conducted study, with a lot of meaningful results, however, some points need to be improved, clarified.  

 INTRODUCTION:

If the introduction is clear and concise, a paragraph on the choice of a diabetic fracture model should be introduced and explained. It seems that choosing this model will add to the complexity of the study. In a fracture model, choosing an immunocompetent mice strain to investigate the anti-inflammatory, and non-immunogenicity of the MSCs, as well as their resulting EVs seems appropriate. I am just not sure why the model would need to be diabetic in addition. The authors certainly have a good reason for that, this should be explained.

 Materials and Methods:

This section is in general clear except for a very important point in my point of view. (§2.1). Human MSC are isolated, then expended in an apparently self-developed serum-free medium (regular amplification medium) until P3 then frozen.

 P3 cells are used as positive control cells. Some other cells are amplified/cultured until P6 (P6hBM-MSC called regular-P6 in this review). A 3 cell population is also studied: regular-P6 cells subjected to additional 70 days of culture without any media changes (starved-P6).

 For some analysis e.g. Flow cytometry, Regular-P3, regular-P6 and starved-P6 are compared. For the other analysis (including in vivo) Regular-P3 and Starved-P6 are compared: why not use Regular-P6 in this case?

This does not seem very clear, or not logical. Using regular-P6 would decrease cell-aging-related differences between the groups, or did I understand wrong? In this case, it should be better explained.

In addition, cells are all in a serum-free medium with some extra growth factors cocktails (only patent form is referenced) a short description of this would be useful for possible repetition of this method.

 §2.9 Fracture model. Please explain why diabetic mice were developed and used and not wild-type animals?

 Results:

In general, a lot for results are presented, sometimes in a redundant manner, making the figure sometimes difficult to follow.

 Specifically:

Figure 1.

(A)A good scheme, but "70days no media changes" does correspond to "starvation", if yes, please group the words better, if not, please be clearer in the corresponding Mat & meth (on the graph it looks like the starvation starting point is at day 70 and not during 70 days.

(B) scale bars are difficult to read

(C) CD146 also seems to be differently expressed in the 3 cell groups. This was not further discussed.

Figure 2:

(D) I guess this graph represents the alizarin red quantification. If yes please say it, if not please clarify.

 Figure 3 and corresponding §3.2:

Referred picture are wrong in the text (Fig 1E should be Fig 3E ).

 Figure4 and § 3.3.

As mentioned before the specific choice of a diabetic mice model should be explained, as this does not seem necessary to me. But I might be wrong, so please explain.

In addition, the text refers to the wrong figure numbers.

This figure is overloaded and is difficult to follow as the letters are quite difficult to allocate to a specific graph. In addition (L) is another view and a combination of (J) and (K), feels redundant.

Figure 5. Good

Figure 6 and related §.

Figure 6 wrongly reference in the text (Figure 5B-E).

(A to E). The purple/blue background overlaps with the light purple of the callus on the CT 3D reconstruction. Difficult to distinguish.

(G) if it gives a clear view of what is happening, this is another representation of F, but its legend mixes up with Picture below. As well (F) legend mixes up with (G).

In all other graphs borders should be clear as it kind of all mixes up..

 Figure 7 and corresponding text.

 See comments for figure 6 (color of the reconstruction etc). In addition (F) is the same type of graph as in 6(F) but no homogeneity. As well in general graphs styles should be homogenized between figures, and within figures.

 Discussion:

Some other extra points could be discussed.

e.g. possible influence of waste metabolites in the starved P6 conditions.

e.g. Possible translation of this knowledge in the clinic, as one of the main concerns about using stem cells in clinic is the need of cell amplification outside the OR. 

e.g. would the research on MSC therapeutic potential focus more EVs? condition media/secretome? 

Author Response

Dear Reviewer 1,

The authors truly appreciate your comments and feedback on our manuscript, and the present form of the manuscript has gained appeal. To facilitate the review process, we have organized this rebuttal by addressing Reviewers’ comments (shown in red) point-by-point and in the same order they were presented.

Comments and Suggestions for Authors

The authors present a very complete study on the possible interest of pre-conditioning mesenchymal stem cells in stringent conditions prior to implantation. The authors hypothesis that promoting cell quiescence would potentially results in better cellular adaptation upon in vivo implantation in the context of bone fracture healing, with emphasis on their anti-inflammatory and regenerative properties.  

In the first part of their study, the authors focused on the in-vitro characterization of starved MSCs compared to classically cultured MSC in serum free medium (called regular MSC in this report). Deep analysis of surfaces markers, and cytokine production were performed, as well as tri-lineage differentiation capacity of the starved cells versus regular cells, or versus extra-cellular vesicles isolated from regular-cells conditioned media.

In the 2nd part of this work, the authors developed/used a diabetic fracture repair model in mice (C57BL/6J) for the study the in-vivo efficiency of cells / conditioned media prepared as described in the first part on bone healing and repair.     

 In general, this is a nicely designed and conducted study, with a lot of meaningful results, however, some points need to be improved, clarified.  

Re: We would like to thank the Reviewer for His/Her accurate evaluation of our original submission and revised the manuscript accordingly as outlined below.

INTRODUCTION:

If the introduction is clear and concise, a paragraph on the choice of a diabetic fracture model should be introduced and explained. It seems that choosing this model will add to the complexity of the study. In a fracture model, choosing an immunocompetent mice strain to investigate the anti-inflammatory, and non-immunogenicity of the MSCs, as well as their resulting EVs seems appropriate. I am just not sure why the model would need to be diabetic in addition. The authors certainly have a good reason for that, this should be explained.

Re: Thanks to the Reviewer's incisive suggestion, the manuscript now includes a new paragraph which discusses why we chose a diabetic model, which is reported as follows (lines 85-94; page 2):

Diabetes is a disease that triggers chronic-hyper-inflammatory conditions [1–3,4], impairing the activity and presence of neutrophils [1], monocytes, and dendritic cells [5] that result in pro-inflammatory and chemotactic mediators, such as IL-6, MCP-1 (CCL2), IL-1, TNF-α, IL-4, and IFN-γ persistent secretion [6–9,10]. All those conditions contribute to induce anti-osteogenic activity, decreased soft callus formation, and enhanced osteoclastic and bone turnover activity [11], thereby fracture healing delay [1,3,12].

Such a hyper-inflammatory milieu is recapitulated in the diabetic STZ-induced animal models used in this research, as well as by McCauley J. et al. [13], and it lays at the basis of the reason why we chose this fracture model to test our hypothesis.

Materials and Methods:

This section is in general clear except for a very important point in my point of view. (§2.1). Human MSC are isolated, then expended in an apparently self-developed serum-free medium (regular amplification medium) until P3 then frozen.

P3 cells are used as positive control cells. Some other cells are amplified/cultured until P6 (P6hBM-MSC called regular-P6 in this review). A 3 cell population is also studied: regular-P6 cells subjected to additional 70 days of culture without any media changes (starved-P6).

For some analysis e.g. Flow cytometry, Regular-P3, regular-P6 and starved-P6 are compared. For the other analysis (including in vivo) Regular-P3 and Starved-P6 are compared: why not use Regular-P6 in this case?

This does not seem very clear, or not logical. Using regular-P6 would decrease cell-aging-related differences between the groups, or did I understand wrong? In this case, it should be better explained.

Re: The authors thank the Reviewer for raising this point. Due to the necessity to obtain such a large number of cells (about 11 x 106) in order to be injected and perform the characterization experiments (Flow cytometry, differentiation assays), we were not able to use them at lower passage (P3). In addition, even though we were aware of the fact that using cells at the same passage (P6) would have reduced the aging-related variability, we decided to test the cells at different passages also to increase the relevance of our results by using cells slightly older and potentially with reduced stemness properties, as seen by [14]. In fact, it is widely known that as passages increase, in vitro cultured stem cells tend to age, losing their stemness and differentiation capacity. From this perspective, this experimental strategy was adopted to maximize the value of the preconditioning technique.

In addition, cells are all in a serum-free medium with some extra growth factors cocktails (only patent form is referenced) a short description of this would be useful for possible repetition of this method.

Re: Media composition is consultable on the patent site https://patents.google.com/patent/WO2015121471A1, however in this revised version of the manuscript we also reported its  formulation (lines 108-114 page 3), α-minimum essential medium (α-MEM, Gibco, Grand Island, NY, USA), lipoprotein 40μg (Sigma-Aldrich, Saint Louis, MO, USA), dexamethasone 50nM (Sigma-Aldrich), ascorbic acid 2-phosphate 100μM (Sigma-Aldrich), human serum albumin 1% (Baxter, Deerfield, IL, USA), insulin-transferrin-selenium 1% (ITS, BD Biosciences, Oxford, CA, USA), transforming growth factor β (TGF-β) 20 ng/ml and fibroblast growth factor 2 (FGF-2) 10 ng/ml.

  • 2.9 Fracture model. Please explain why diabetic mice were developed and used and not wild-type animals?

Re: Please see first response on the model.

Results:

In general, a lot for results are presented, sometimes in a redundant manner, making the figure sometimes difficult to follow.

Re: We have revised the article to present our results in a more concise manner.  

Specifically:

Figure 1.

(A)A good scheme, but "70days no media changes" does correspond to "starvation", if yes, please group the words better, if not, please be clearer in the corresponding Mat & meth (on the graph it looks like the starvation starting point is at day 70 and not during 70 days.

Re: Thank you for suggesting this improvement to the graph. The words “no media change” in the graph has been clearly associated with the starvation process.

(B) scale bars are difficult to read

Re: Thank you for drawing attention to this issue. The size of the scale bars has been increased to add emphasis to them.

(C) CD146 also seems to be differently expressed in the 3 cell groups. This was not further discussed.

Re: We are thankful for this Reviewer’s criticism and we have added a new paragraph discussing the reduced expression of CD-146. The paragraph is reported in the text as follows (lines 612-618; page 18): CD146 reduction is not a surprising event that was reported to be dependent on the culture conditions, aging and leads to reduced osteoblastic activity. Since our culture medium is composed of α-MEM, which was not seen to cause CD-146 reduction, even at higher passages (P8)[14], suggests that the starvation process is the main cause of both its decrease and, as a consequence, reduced hBM-MSC osteoblastic potential. On the contrary, comparison of the results obtained from P6 starved cells and P3 regularly fed cells showed no reduction of their osteoblastic capability for P6 starved cells[14].

Figure 2:

(D) I guess this graph represents the alizarin red quantification. If yes please say it, if not please clarify.

Re: Figure 2D represents the summary graph that compares the three differentiation assays used to confirm the osteoblastic, chondroblastic, and adipocytic differentiation potential, shown as a ratio with respect to undifferentiated control cells at the same passage. The figure caption have been also improved and the text reported as follows (lines 322-328 page 9): Starved cells retain their ability to differentiate and survive secreting trophic factors: (A) Oil red O staining comparing the hBM-MSC adipocytic differentiation potential in regularly fed and starved cell populations. (B) Alizarin red staining showing the osteoblastic potential of the tested cells in the same groups of A. (C) Alcian blue and safranin O staining evidencing the chondroblastic potential of the starved and regularly fed cells. (D) Whisker plot comparing the differentiation potential of the tested hBM-MSC. Results are normalized with respect to control cells cultured in proliferation medium. Results are presented as mean ± SD of three independent experiments. *p ≤ 0.05 (one-way ANOVA followed by Bonferroni's post hoc test).

Figure 3 and corresponding §3.2:

Referred picture are wrong in the text (Fig 1E should be Fig 3E).

Re: We apologize for the incorrect figures that were cited in the text. The referred images have now been corrected.

Figure4 and § 3.3.

As mentioned before the specific choice of a diabetic mice model should be explained, as this does not seem necessary to me. But I might be wrong, so please explain.

Re: Please see first response on the model.

In addition, the text refers to the wrong figure numbers.

Re: We appreciate the Reviewer pointing out the error. References to the figures have been modified.

This figure is overloaded and is difficult to follow as the letters are quite difficult to allocate to a specific graph. In addition (L) is another view and a combination of (J) and (K), feels redundant.

Re: Thanks again, the letters and the whole figure have been edited to improve the overall clarity. Authors tried to simplify readers' comprehension, even adding a summarizing graph (L) due to the fact that the higher number of cytokines which were not completely understandable with merely the box whisker plots (J, K).

Figure 5. Good

Figure 6 and related §.

Figure 6 wrongly reference in the text (Figure 5B-E).

Re: We appreciate the Reviewer pointing out the error. References to the figure have been modified.

(A to E). The purple/blue background overlaps with the light purple of the callus on the CT 3D reconstruction. Difficult to distinguish.

Re: Following the keen Reviewer’s advice, the background colour has been changed to yellow. Now the newly formed callus (blue) should be distinguishable from the bone and the background.

(G) if it gives a clear view of what is happening, this is another representation of F, but its legend mixes up with Picture below. As well (F) legend mixes up with (G).

In all other graphs borders should be clear as it kind of all mixes up..

Re: Graphs have been reorganized and they should not overlay now. Thanks to the Reviewer, we believe that the figures are now much clearer than previously.

Figure 7 and corresponding text.

See comments for figure 6 (color of the reconstruction etc). In addition (F) is the same type of graph as in 6(F) but no homogeneity. As well in general graphs styles should be homogenized between figures, and within figures.

Re: Graphs should no longer overlap as a result of the reorganization. The Reviewer's comments have made the figures easier to follow.

Discussion:

Some other extra points could be discussed.

e.g. possible influence of waste metabolites in the starved P6 conditions.

Re: According to the Reviewer's request, the following topics have been covered in the revised manuscript (lines 599-611; page 18):

In our study, we subjected our cells to an intense preconditioning procedure, which, as demonstrated in our previously published paper, results in a significant reduction in the number of cells during starvation and may increase the presence of waste/toxic molecules like lactate and ammonia[15]. This likely contributed to increasing the osmolarity and ionic strength of the culture medium[16]. Although we did not measure the level of ammonia in our previous research, we found that lactate transiently increased in the culture medium until day 27 before decreasing towards the end of the starving period[16]. Recent research demonstrating that lactate supplementation can decrease both lactate and ammonia levels in culture[17]. These results, along with the presence of albumin, both as messenger and protein, in starvation-induced conditions [16], which is known to be a lipid and metal ion transporter with antioxidant and buffering properties, imply that the starvation process activates specific cellular adaptive mechanisms that lessens the negative effects.

e.g. Possible translation of this knowledge in the clinic, as one of the main concerns about using stem cells in clinic is the need of cell amplification outside the OR. e.g. would the research on MSC therapeutic potential focus more EVs? condition media/secretome? 

Re: The requested points have been discussed in the new version of the manuscript as requested by the Reviewer and reported as follows (lines 715-725; page 20): Since the main concern about using stem cells in clinical settings is the need for cell expansion outside their quiescent niches, we previously suggested in our published research [17] that MSC culture in xenogen-free medium would be the preferred choice for future MSC in vitro amplification due to its reduced nutritional content. In addition, we suggest that deep studies of the starvation approach at different time points, both in normoxia or hypoxia, and for different periods, will also reveal the survival, regenerative and anti-inflammatory properties of the cells. Importantly, the therapeutic potential of the MSC has to be explored also by testing the properties of the EVs retrieved from starved cells.

Reviewer 2 Report

In their study "Regenerative and anti-inflammatory potential of regularly fed,  starved cells and extracellular vesicles in vivo" Mr Ferro and co-workers have addressed an treatment of MSC (starved and regulary fed) on a fracture healing model which contains a mouse diabetes model. This is quite a translational set-up and the combination of experiments is very special. However the study has some issues.

Major: Especially because you discuss quite a lot with cell types and soft tissue, the story would befit from histological experiments quite a lot

Many figures are quite confusing. I would ask you to review every figure, if it is possible, if it is clear. I will try to address it in the minor issues, but I am quite sure that I have missed some

Please write at each experiment which statistical test you have used in that case.

Minor: Why did you use starved cells in P6 and regulary fed in P3?

L212 which method is behind the qiagen multiplex analysis? ELISA?

Figure 1c The percent are a ratio of what in paricular?

l291 how do you calculate the Arbitrary units? If these are mesh ups of score you cannot do statistical analyses. Are the AU of Figure 2d the same as in 2e?

Figure 3a please mark the structures you identify as EV

Figure 3c are the EVs in starved MSCs different in the protein production on western blot?

L342 The reference on the images is absolutly chaotic, please double check the right cross references

Figure 4d-g the X-rays are not labelled

Figure 4l, what is the label exactly? (x-fold increased protein concentration?)

Figure 5 you switch the styles of your graph so it is not really possible to compare Expression and multiplex analyses. Please use similar graphs

l450 like above the cross references to the figures is not right

Figure 6 Light purple is not a good color for newly formed callus when you use violett as a background color!

Figure 6g and 7f this representation is not really good if you use parameters indication a positive bone phenotype like BV/TV or trabecular thickness with parameters standing for a negative bone phenotype like trabecular separation

Figure 4 showed just the relative histomorphometric parameters from the µCT, Figure 6 and 7 showed the relative and absolute data. I would suggest to put all the graphs with the absolute data in the supplemental data part. 

Author Response

Dear Reviewer 2,

We much appreciated your critical feedback on our article, and we believe this revised version is now much more understandable and complete. We have organized this letter to simplify the review process by addressing Reviewers' comments (shown in red) point-by-point and in the same order they were made.

Comments and Suggestions for Authors

In their study "Regenerative and anti-inflammatory potential of regularly fed, starved cells and extracellular vesicles in vivo" Mr Ferro and co-workers have addressed an treatment of MSC (starved and regulary fed) on a fracture healing model which contains a mouse diabetes model. This is quite a translational set-up and the combination of experiments is very special. However the study has some issues.

Re: We would like to thank the Reviewer for His/Her positive evaluation of our manuscript and revised the manuscript accordingly to address all the raised issues.

Major: Especially because you discuss quite a lot with cell types and soft tissue, the story would befit from histological experiments quite a lot

Re: We agree with the Reviewer, that the study could benefit from histological experiments. We acknowledge that this is a main limitation of the study, and we have reported it in the discussion (lines 693-694, pages 20). Indeed, we are going to perform such analysis in future in vivo studies. The text is reported as follows: Further studies based on histological analysis will be needed to fully understand the response to the proposed cell therapy.

Many figures are quite confusing. I would ask you to review every figure, if it is possible, if it is clear. I will try to address it in the minor issues, but I am quite sure that I have missed some.

Re: We have revised the figures and thanks to the Reviewer's comments they are now much easier to follow.

Please write at each experiment which statistical test you have used in that case.

Re: We thank the Reviewer a lot for the advice. Statistical tests have been added to all the figure captions.

Minor: Why did you use starved cells in P6 and regularly fed in P3?

Re: As highlighted by both Reviewers 1 and 2, given the large number of cells (about 11 X106) to be injected and to perform the characterization experiments (Flow cytometry, differentiation assays), we were not able to use them at lower passages. In addition, even though we are aware of the fact that using cells at the same passage would have reduced the aging-related variability, we decided to test the cells at different passages in order to obtain the necessary number of injectable cells as well as increase the relevance of our results by using cells slightly older and with less stemness properties, as seen by [14].

In fact, it is widely known that as passages increase, in vitro cultured stem cells tend to age, losing their stemness and differentiation capacity. From this perspective, this experimental strategy was adopted to maximize the value of the preconditioning technique.

L212 which method is behind the qiagen multiplex analysis? ELISA?

Re: Yes, the assay is based on the enzyme-linked immunosorbent assay (ELISA), and now specified materials and methods.

Figure 1c The percent are a ratio of what in paricular?

Re: We appreciate you pointed out this omitted reference. The expression percentage was obtained respect total cell count 2 x 105 and the related figure 1C was updated.

l291 how do you calculate the Arbitrary units? If these are mesh ups of score you cannot do statistical analyses. Are the AU of Figure 2d the same as in 2e?

Re: They are expressed as a ratio to undifferentiated controls. The Y axis of Figure 2D and E have been renamed to fold change with respect to undifferentiated or fresh medium controls.

Figure 3a please mark the structures you identify as EV

Re: We marked the EVs with red circles as the Reviewer had requested.

Figure 3c are the EVs in starved MSCs different in the protein production on western blot?

Re: The EVs from starving cells were not tested and this might be indeed the following step. In fact, we argue (in lines 719-721 page20) that in order to fully understand the therapeutic potential of MSC, it is necessary to examine the characteristics of EVs recovered from starved cells.

L342 The reference on the images is absolutly chaotic, please double check the right cross references

Re: The right cross references have been improved by having their dimension reduced thanks to the Reviewer.

Figure 4d-g the X-rays are not labelled

Re: The text and the figure legend now explicitly specify which images are referenced. The text is reported in (line 373-376 and page 11) as it follows “X-ray scans taken immediately after the creation of the mice's femur transverse fractures (Fig. 4D) and those taken two days later (Fig. 4E) directed the procedure for injecting cells into the fracture site. Additionally, X-rays were used to track the healing of fractures at days seven (Fig. 4F) and 23 (Fig. 4G)” and the figure legend, (line 394-396 and page 12) and, is reported as follows “(D-G) X-ray scans of the mouse femur immediately after fracture (D), after two days (E), which directed the approach for treatment into the fracture site, and after 7 (F) and 23 days (G)”.

Figure 4l, what is the label exactly? (x-fold increased protein concentration?)

Re: Yes, it is fold change to fractured-only and now it has been specified in the figure legend.

Figure 5 you switch the styles of your graph so it is not really possible to compare Expression and multiplex analyses. Please use similar graphs

Re: Due to the large number of tested markers, we first presented the measured values using a box whisker graph style, and then we switched to radar graphs because, in our opinion, they aid to simplify the findings and enhance readers' comprehension.

l450 like above the cross references to the figures is not right

Re: As suggested before, right cross references have been improved by reducing their dimension

Figure 6 Light purple is not a good color for newly formed callus when you use violett as a background color!

Re: Following the Reviewer’s observation, the background colour has been changed to yellow. Now the newly formed callus (blue) should be distinguishable from the bone and the background.

Figure 6g and 7f this representation is not really good if you use parameters indication a positive bone phenotype like BV/TV or trabecular thickness with parameters standing for a negative bone phenotype like trabecular separation

Re: We thank the Reviewer for His/Her comment. Though, to show a more detailed picture of the process, we provided positive and negative indicators together.

Figure 4 showed just the relative histomorphometric parameters from the µCT, Figure 6 and 7 showed the relative and absolute data. I would suggest to put all the graphs with the absolute data in the supplemental data part. 

Re: We agree with the Reviewer suggestion and in the revised paper the relative graphs with the absolute data have been moved to supplementary results and renamed as Supp. Fig 1 and 2.

Supp. Fig. 1. Analysis of the newly formed mineralized tissue five days after cells (starved and regularly fed) and EVs injections at the fracture site. (A-D) With respect to the diabetic fractured mice, preconditioned/starved cells induced a significant improvement in BV/TV, connectivity, Conn.D (mm^-3) and BS (mm2). (E-H) No significant difference was seen between the treatments for trabecular Tb. Th. Mean (mm), Tb. Th. Max (mm), Tb. Sp. Mean (mm) and Tb. Sp. Max. Results are presented as mean ± SD. p≤0.05 *respect EVs, +Cells, **Diabetic fractured (Sample size four to five animals/group, one-way ANOVA followed by Fisher’s post hoc test). Abbreviations: regularly fed cells (Cells); starved cells (Starved); extracellular vesicles (EVs); bone volume/total volume (BV/TV); connectivity density (Conn.D (mm^-3)); bone surface (BS(mm2)); trabecular thickness mean (Tb. Th. Mean (mm)); trabecular thickness max (Tb. Th. Max (mm)); trabecular spacing mean (Tb. Sp. Mean (mm)); trabecular spacing max (Tb. Sp. Max (mm)).

Supp. Fig. 2. Analysis of the newly formed mineralized tissue at 21 days after cell and EVs injection at the fracture site. (A-F) With respect to the diabetic fractured mice, preconditioned/starved cells induced no significant improvement in BV/TV, connectivity, Conn.D (mm^-3) and BS (mm2), trabecular Tb. Th. Mean (mm), Tb. Th. Max (mm) at the later time-point (23 days) as evident by µCT parameters. The results are presented as mean ± SD. (Sample size five to eight animals/group, one-way ANOVA followed by Fisher’s post hoc test). Abbreviations: regularly fed cells (Cells); starved cells (Starved); extracellular vesicles (EVs); bone volume/total volume (BV/TV); connectivity density (Conn.D (mm^-3)); bone surface (BS(mm2)); trabecular thickness mean (Tb. Th. Mean (mm)); (p≤0.05); trabecular thickness max (Tb. Th. Max (mm)); trabecular spacing mean (Tb. Sp. Mean (mm)); trabecular spacing max (Tb. Sp. Max (mm)).

References

(1)        Yan, W.; Li, X. Impact of Diabetes and Its Treatments on Skeletal Diseases. Front Med 2013, 7 (1), 81–90. https://doi.org/10.1007/s11684-013-0243-9.

(2)        Watson, L.; Chen, X. Z.; Ryan, A. E.; Fleming, Á.; Carbin, A.; O’Flynn, L.; Loftus, P. G.; Horan, E.; Connolly, D.; McDonnell, P.; McNamara, L. M.; O’Brien, T.; Coleman, C. M. Administration of Human Non-Diabetic Mesenchymal Stromal Cells to a Murine Model of Diabetic Fracture Repair: A Pilot Study. Cells 2020, 9 (6), 1394. https://doi.org/10.3390/cells9061394.

(3)        Motyl, K.; McCabe, L. R. Streptozotocin, Type I Diabetes Severity and Bone. Biol Proced Online 2009, 11, 296–315. https://doi.org/10.1007/s12575-009-9000-5.

(4)        Davey, G. C.; Patil, S. B.; O’Loughlin, A.; O’Brien, T. Mesenchymal Stem Cell-Based Treatment for Microvascular and Secondary Complications of Diabetes Mellitus. Front Endocrinol (Lausanne) 2014, 5, 86. https://doi.org/10.3389/fendo.2014.00086.

(5)        Maruyama, M.; Rhee, C.; Utsunomiya, T.; Zhang, N.; Ueno, M.; Yao, Z.; Goodman, S. B. Modulation of the Inflammatory Response and Bone Healing. Front Endocrinol (Lausanne) 2020, 11, 386. https://doi.org/10.3389/fendo.2020.00386.

(6)        Loi, F.; Córdova, L. A.; Pajarinen, J.; Lin, T.; Yao, Z.; Goodman, S. B. Inflammation, Fracture and Bone Repair. Bone 2016, 86, 119–130. https://doi.org/10.1016/j.bone.2016.02.020.

(7)        Lacey, D. C.; Simmons, P. J.; Graves, S. E.; Hamilton, J. A. Proinflammatory Cytokines Inhibit Osteogenic Differentiation from Stem Cells: Implications for Bone Repair during Inflammation. Osteoarthritis Cartilage 2009, 17 (6), 735–742. https://doi.org/10.1016/j.joca.2008.11.011.

(8)        Kayal, R. A.; Siqueira, M.; Alblowi, J.; McLean, J.; Krothapalli, N.; Faibish, D.; Einhorn, T. A.; Gerstenfeld, L. C.; Graves, D. T. TNF-Alpha Mediates Diabetes-Enhanced Chondrocyte Apoptosis during Fracture Healing and Stimulates Chondrocyte Apoptosis through FOXO1. J Bone Miner Res 2010, 25 (7), 1604–1615. https://doi.org/10.1002/jbmr.59.

(9)        Liu, Y.; Wang, L.; Kikuiri, T.; Akiyama, K.; Chen, C.; Xu, X.; Yang, R.; Chen, W.; Wang, S.; Shi, S. Mesenchymal Stem Cell-Based Tissue Regeneration Is Governed by Recipient T Lymphocytes via IFN-γ and TNF-α. Nat Med 2011, 17 (12), 1594–1601. https://doi.org/10.1038/nm.2542.

(10)      Deshpande, S.; James, A. W.; Blough, J.; Donneys, A.; Wang, S. C.; Cederna, P. S.; Buchman, S. R.; Levi, B. Reconciling the Effects of Inflammatory Cytokines on Mesenchymal Cell Osteogenic Differentiation. J Surg Res 2013, 185 (1), 278–285. https://doi.org/10.1016/j.jss.2013.06.063.

(11)      Kayal, R. A.; Tsatsas, D.; Bauer, M. A.; Allen, B.; Al-Sebaei, M. O.; Kakar, S.; Leone, C. W.; Morgan, E. F.; Gerstenfeld, L. C.; Einhorn, T. A.; Graves, D. T. Diminished Bone Formation during Diabetic Fracture Healing Is Related to the Premature Resorption of Cartilage Associated with Increased Osteoclast Activity. J Bone Miner Res 2007, 22 (4), 560–568. https://doi.org/10.1359/jbmr.070115.

(12)      Ogasawara, A.; Nakajima, A.; Nakajima, F.; Goto, K.-I.; Yamazaki, M. Molecular Basis for Affected Cartilage Formation and Bone Union in Fracture Healing of the Streptozotocin-Induced Diabetic Rat. Bone 2008, 43 (5), 832–839. https://doi.org/10.1016/j.bone.2008.07.246.

(13)      McCauley, J.; Bitsaktsis, C.; Cottrell, J. Macrophage Subtype and Cytokine Expression Characterization during the Acute Inflammatory Phase of Mouse Bone Fracture Repair. J Orthop Res 2020, 38 (8), 1693–1702. https://doi.org/10.1002/jor.24603.

(14)      Yang, Y.-H. K.; Ogando, C. R.; Wang See, C.; Chang, T.-Y.; Barabino, G. A. Changes in Phenotype and Differentiation Potential of Human Mesenchymal Stem Cells Aging in Vitro. Stem Cell Res Ther 2018, 9 (1), 131. https://doi.org/10.1186/s13287-018-0876-3.

(15)      Schumpp, B.; Schlaeger, E. J. Growth Study of Lactate and Ammonia Double-Resistant Clones of HL-60 Cells. Cytotechnology 1992, 8 (1), 39–44. https://doi.org/10.1007/BF02540028.

(16)      Ferro, F.; Spelat, R.; Shaw, G.; Duffy, N.; Islam, M. N.; O’Shea, P. M.; O’Toole, D.; Howard, L.; Murphy, J. M. Survival/Adaptation of Bone Marrow-Derived Mesenchymal Stem Cells After Long-Term Starvation Through Selective Processes. Stem Cells 2019, 37 (6), 813–827. https://doi.org/10.1002/stem.2998.

(17)      Freund, N. W.; Croughan, M. S. A Simple Method to Reduce Both Lactic Acid and Ammonium Production in Industrial Animal Cell Culture. Int J Mol Sci 2018, 19 (2). https://doi.org/10.3390/ijms19020385.

(18)      Bahney, C. S.; Zondervan, R. L.; Allison, P.; Theologis, A.; Ashley, J. W.; Ahn, J.; Miclau, T.; Marcucio, R. S.; Hankenson, K. D. Cellular Biology of Fracture Healing. J Orthop Res 2019, 37 (1), 35–50. https://doi.org/10.1002/jor.24170.

(19)      Nam, D.; Mau, E.; Wang, Y.; Wright, D.; Silkstone, D.; Whetstone, H.; Whyne, C.; Alman, B. T-Lymphocytes Enable Osteoblast Maturation via IL-17F during the Early Phase of Fracture Repair. PLoS One 2012, 7 (6), e40044. https://doi.org/10.1371/journal.pone.0040044.

(20)      Gerstenfeld, L. C.; Cho, T. J.; Kon, T.; Aizawa, T.; Tsay, A.; Fitch, J.; Barnes, G. L.; Graves, D. T.; Einhorn, T. A. Impaired Fracture Healing in the Absence of TNF-Alpha Signaling: The Role of TNF-Alpha in Endochondral Cartilage Resorption. J Bone Miner Res 2003, 18 (9), 1584–1592. https://doi.org/10.1359/jbmr.2003.18.9.1584.

(21)      Vries, M. H. M.; Wagenaar, A.; Verbruggen, S. E. L.; Molin, D. G. M.; Dijkgraaf, I.; Hackeng, T. H.; Post, M. J. CXCL1 Promotes Arteriogenesis through Enhanced Monocyte Recruitment into the Peri-Collateral Space. Angiogenesis 2015, 18 (2), 163–171. https://doi.org/10.1007/s10456-014-9454-1.

(22)      Gerstenfeld, L. C.; Cho, T. J.; Kon, T.; Aizawa, T.; Cruceta, J.; Graves, B. D.; Einhorn, T. A. Impaired Intramembranous Bone Formation during Bone Repair in the Absence of Tumor Necrosis Factor-Alpha Signaling. Cells Tissues Organs 2001, 169 (3), 285–294. https://doi.org/10.1159/000047893.

Reviewer 3 Report

The manuscript is written clearly and put together in an easy to follow, logical manner. The experimental approach makes good sense and is sufficiently thorough. And the conclusion is supported by the data.

Recommendations are shown below.

1. Please state the sample size of each group in figure legend. In fig. 4(B-C,I-K), the sample size of HBSS group,Fractured 23 DAYs group and  Diabetic Fractured group is suggested to be stated clearly.Although it's said that sample size four to five animals/group in figure legend, it seems only 2-3 samples in those group mentioned above;
2.  Line  523-524, "osteoclastic  and bone turnover activity" is suggested to be replace with "osteoclastic activity and bone resorption ";
3. Incorrect punctuation marks and formatting errors exist in the manuscript. e.g. line 538, line 543-544, line 605, ...;
4. It's suggested to add a figure legend for figure 8;
5. Lots of inflammatory cytokines were tested in this study and their roles in the healing process of fracture need to be further discussed in the discussion.

Author Response

Dear Reviewer 3,

We greatly valued the Reviewer's comments and recommendations, and we believe they helped the manuscript's quality in this amended version. We have organized this letter to simplify the review process by addressing Reviewers' comments (shown in red) point-by-point and in the same order they were made.

Comments and Suggestions for Authors

The manuscript is written clearly and put together in an easy to follow, logical manner. The experimental approach makes good sense and is sufficiently thorough. And the conclusion is supported by the data.
Re: We thank the Reviewer for His/Her strong support to our paper.

Recommendations are shown below.

1. Please state the sample size of each group in figure legend. In fig. 4(B-C,I-K), the sample size of HBSS group,Fractured 23 DAYs group and  Diabetic Fractured group is suggested to be stated clearly.Although it's said that sample size four to five animals/group in figure legend, it seems only 2-3 samples in those group mentioned above;

Re: Following the Reviewer's suggestion, which is in line with Reviewer 2, all the animals per group specification have been added to the relative figures in the figure caption.

  1. Line  523-524, "osteoclastic  and bone turnover activity" is suggested to be replace with "osteoclastic activity and bone resorption ";

Re: We agree with the Reviewer's suggestion, and the wording "osteoclastic  and bone turnover activity" has been changed to “osteoclastic activity and bone resorption”  (line 587 page 17).

  1. Incorrect punctuation marks and formatting errors exist in the manuscript. e.g. line 538, line 543-544, line 605,...;

Re: Thank you; the manuscript has been thoroughly reviewed and edited to remove any misspelled words and incorrect formatting.

4. It's suggested to add a figure legend for figure 8;

Re: We agree on the fact that a more detailed figure caption was necessary for figure 8. Consequently, a more informative figure legend has been added to clearly understand the figure. The new caption has been reported as follows (lines 674-681 page 19-20): Results demonstrated that the starvation process induces the production of different growth factors such as IGFBP2, 3, 6, HGF, TGFβ2, PDGFα, VEGFR2 and IGF2 which induce the secretion of cytokines such as IL-17 and repress the production of IL-1α, β, 2, 10 and 13 at the earliest time-point. At day 7 after fracture, starved cells favor the expression of IL-6, 13 and KC, increase the presence of mineralized matrix, and presumably the soft callus formation at the fracture site, with a decrease in the expression of IL-1α, 12(p70),17,GM-CSF, IFN-γ, MIP-1α, β and RANTES. However, at 23 days post-fracture, no difference was seen within the tested conditions.

  1. Lots of inflammatory cytokines were tested in this study and their roles in the healing process of fracture need to be further discussed in the discussion.

Re: We are thankful to the Reviewer for having suggested to add a new paragraph regarding the influence of the inflammatory cytokines during fracture healing. Thus, a new paragraph was added and reported as follows (lines 565-583 page 17):  

Inflammatory cells produce a variety of cytokines, influencing in both positive and negative ways cells and the various phases of bone regeneration [6,18].

In fact, neutrophils release pro-inflammatory and chemotactic mediators like IL-6 and MCP-1 (CCL2)[6] during the initial step. Monocyte/macrophages secrete as well many different cytokines such as interleukin-1α and β (IL-1α and β) which suppress matrix mineralization and osteoblastic differentiation [7]. TNF-α reduces chondroblasts viability [8], inhibits osteogenic differentiation[7] and reduces bone formation in mice [9]. IL-4 and interferon γ (IFN-γ) are also significantly antiosteogenic[10]. Moreover, IFN-γ and TNF-α synergistically induce BM-MSC apoptosis and significantly inhibit bone formation in vivo[9], while IL-17 stimulates the nuclear factor kappa B (NF-κB) signalling pathway and impairs the differentiation of BM-MSCs[18].

At later stages many different studies also demonstrated that inflammatory and chemotactic mediators such as TNF-α, IL-4, IL-6, IL-13, IFN-γ, KC and MCP-1 released from macrophages, lymphocytes and eosinophils, stimulate also the recruitment of fibroblasts, mesenchymal stem cells (MSCs), and osteoprogenitor cells from their niches [6,19,20,21]. For example, it was seen that mice missing the TNF-α receptor gene have a significant delay in the chondrocyte differentiation [20], and endochondral ossification [22]. Interleukin 6 (IL6) has been similarly implicated in bone healing delay as well as its lack with reduced fracture healing [13].

References

(1)        Yan, W.; Li, X. Impact of Diabetes and Its Treatments on Skeletal Diseases. Front Med 2013, 7 (1), 81–90. https://doi.org/10.1007/s11684-013-0243-9.

(2)        Watson, L.; Chen, X. Z.; Ryan, A. E.; Fleming, Á.; Carbin, A.; O’Flynn, L.; Loftus, P. G.; Horan, E.; Connolly, D.; McDonnell, P.; McNamara, L. M.; O’Brien, T.; Coleman, C. M. Administration of Human Non-Diabetic Mesenchymal Stromal Cells to a Murine Model of Diabetic Fracture Repair: A Pilot Study. Cells 2020, 9 (6), 1394. https://doi.org/10.3390/cells9061394.

(3)        Motyl, K.; McCabe, L. R. Streptozotocin, Type I Diabetes Severity and Bone. Biol Proced Online 2009, 11, 296–315. https://doi.org/10.1007/s12575-009-9000-5.

(4)        Davey, G. C.; Patil, S. B.; O’Loughlin, A.; O’Brien, T. Mesenchymal Stem Cell-Based Treatment for Microvascular and Secondary Complications of Diabetes Mellitus. Front Endocrinol (Lausanne) 2014, 5, 86. https://doi.org/10.3389/fendo.2014.00086.

(5)        Maruyama, M.; Rhee, C.; Utsunomiya, T.; Zhang, N.; Ueno, M.; Yao, Z.; Goodman, S. B. Modulation of the Inflammatory Response and Bone Healing. Front Endocrinol (Lausanne) 2020, 11, 386. https://doi.org/10.3389/fendo.2020.00386.

(6)        Loi, F.; Córdova, L. A.; Pajarinen, J.; Lin, T.; Yao, Z.; Goodman, S. B. Inflammation, Fracture and Bone Repair. Bone 2016, 86, 119–130. https://doi.org/10.1016/j.bone.2016.02.020.

(7)        Lacey, D. C.; Simmons, P. J.; Graves, S. E.; Hamilton, J. A. Proinflammatory Cytokines Inhibit Osteogenic Differentiation from Stem Cells: Implications for Bone Repair during Inflammation. Osteoarthritis Cartilage 2009, 17 (6), 735–742. https://doi.org/10.1016/j.joca.2008.11.011.

(8)        Kayal, R. A.; Siqueira, M.; Alblowi, J.; McLean, J.; Krothapalli, N.; Faibish, D.; Einhorn, T. A.; Gerstenfeld, L. C.; Graves, D. T. TNF-Alpha Mediates Diabetes-Enhanced Chondrocyte Apoptosis during Fracture Healing and Stimulates Chondrocyte Apoptosis through FOXO1. J Bone Miner Res 2010, 25 (7), 1604–1615. https://doi.org/10.1002/jbmr.59.

(9)        Liu, Y.; Wang, L.; Kikuiri, T.; Akiyama, K.; Chen, C.; Xu, X.; Yang, R.; Chen, W.; Wang, S.; Shi, S. Mesenchymal Stem Cell-Based Tissue Regeneration Is Governed by Recipient T Lymphocytes via IFN-γ and TNF-α. Nat Med 2011, 17 (12), 1594–1601. https://doi.org/10.1038/nm.2542.

(10)      Deshpande, S.; James, A. W.; Blough, J.; Donneys, A.; Wang, S. C.; Cederna, P. S.; Buchman, S. R.; Levi, B. Reconciling the Effects of Inflammatory Cytokines on Mesenchymal Cell Osteogenic Differentiation. J Surg Res 2013, 185 (1), 278–285. https://doi.org/10.1016/j.jss.2013.06.063.

(11)      Kayal, R. A.; Tsatsas, D.; Bauer, M. A.; Allen, B.; Al-Sebaei, M. O.; Kakar, S.; Leone, C. W.; Morgan, E. F.; Gerstenfeld, L. C.; Einhorn, T. A.; Graves, D. T. Diminished Bone Formation during Diabetic Fracture Healing Is Related to the Premature Resorption of Cartilage Associated with Increased Osteoclast Activity. J Bone Miner Res 2007, 22 (4), 560–568. https://doi.org/10.1359/jbmr.070115.

(12)      Ogasawara, A.; Nakajima, A.; Nakajima, F.; Goto, K.-I.; Yamazaki, M. Molecular Basis for Affected Cartilage Formation and Bone Union in Fracture Healing of the Streptozotocin-Induced Diabetic Rat. Bone 2008, 43 (5), 832–839. https://doi.org/10.1016/j.bone.2008.07.246.

(13)      McCauley, J.; Bitsaktsis, C.; Cottrell, J. Macrophage Subtype and Cytokine Expression Characterization during the Acute Inflammatory Phase of Mouse Bone Fracture Repair. J Orthop Res 2020, 38 (8), 1693–1702. https://doi.org/10.1002/jor.24603.

(14)      Yang, Y.-H. K.; Ogando, C. R.; Wang See, C.; Chang, T.-Y.; Barabino, G. A. Changes in Phenotype and Differentiation Potential of Human Mesenchymal Stem Cells Aging in Vitro. Stem Cell Res Ther 2018, 9 (1), 131. https://doi.org/10.1186/s13287-018-0876-3.

(15)      Schumpp, B.; Schlaeger, E. J. Growth Study of Lactate and Ammonia Double-Resistant Clones of HL-60 Cells. Cytotechnology 1992, 8 (1), 39–44. https://doi.org/10.1007/BF02540028.

(16)      Ferro, F.; Spelat, R.; Shaw, G.; Duffy, N.; Islam, M. N.; O’Shea, P. M.; O’Toole, D.; Howard, L.; Murphy, J. M. Survival/Adaptation of Bone Marrow-Derived Mesenchymal Stem Cells After Long-Term Starvation Through Selective Processes. Stem Cells 2019, 37 (6), 813–827. https://doi.org/10.1002/stem.2998.

(17)      Freund, N. W.; Croughan, M. S. A Simple Method to Reduce Both Lactic Acid and Ammonium Production in Industrial Animal Cell Culture. Int J Mol Sci 2018, 19 (2). https://doi.org/10.3390/ijms19020385.

(18)      Bahney, C. S.; Zondervan, R. L.; Allison, P.; Theologis, A.; Ashley, J. W.; Ahn, J.; Miclau, T.; Marcucio, R. S.; Hankenson, K. D. Cellular Biology of Fracture Healing. J Orthop Res 2019, 37 (1), 35–50. https://doi.org/10.1002/jor.24170.

(19)      Nam, D.; Mau, E.; Wang, Y.; Wright, D.; Silkstone, D.; Whetstone, H.; Whyne, C.; Alman, B. T-Lymphocytes Enable Osteoblast Maturation via IL-17F during the Early Phase of Fracture Repair. PLoS One 2012, 7 (6), e40044. https://doi.org/10.1371/journal.pone.0040044.

(20)      Gerstenfeld, L. C.; Cho, T. J.; Kon, T.; Aizawa, T.; Tsay, A.; Fitch, J.; Barnes, G. L.; Graves, D. T.; Einhorn, T. A. Impaired Fracture Healing in the Absence of TNF-Alpha Signaling: The Role of TNF-Alpha in Endochondral Cartilage Resorption. J Bone Miner Res 2003, 18 (9), 1584–1592. https://doi.org/10.1359/jbmr.2003.18.9.1584.

(21)      Vries, M. H. M.; Wagenaar, A.; Verbruggen, S. E. L.; Molin, D. G. M.; Dijkgraaf, I.; Hackeng, T. H.; Post, M. J. CXCL1 Promotes Arteriogenesis through Enhanced Monocyte Recruitment into the Peri-Collateral Space. Angiogenesis 2015, 18 (2), 163–171. https://doi.org/10.1007/s10456-014-9454-1.

(22)      Gerstenfeld, L. C.; Cho, T. J.; Kon, T.; Aizawa, T.; Cruceta, J.; Graves, B. D.; Einhorn, T. A. Impaired Intramembranous Bone Formation during Bone Repair in the Absence of Tumor Necrosis Factor-Alpha Signaling. Cells Tissues Organs 2001, 169 (3), 285–294. https://doi.org/10.1159/000047893.

Round 2

Reviewer 1 Report

We thank the authors for taking into account our previous comments. 

The article has gained clarity. We now understand why P3 and P6 cells were used, it is well explained. I just still ask myself if using the cells all in P6 might not make more sense.

Reviewer 2 Report

The revision has been done in a satisfactory amount.